# Unifying Diffusion Models' Latent Space, with Applications to CycleDiffusion and Guidance

## Abstract

Diffusion models have achieved unprecedented performance in generative modeling. The commonly-adopted formulation of the latent code of diffusion models is a sequence of gradually denoised samples, as opposed to the simpler (e.g., Gaussian) latent space of GANs, VAEs, and normalizing flows. This paper provides an alternative, Gaussian formulation of the latent space of diffusion models, as well as a reconstructable **DPM-Encoder** that maps images into the latent space. While our formulation is purely based on the definition of diffusion models, we demonstrate several intriguing consequences. (1) Empirically, we observe that a common latent space emerges from two diffusion models trained independently on related domains. In light of this finding, we propose **CycleDiffusion**, which uses DPM-Encoder for unpaired image-to-image translation. Furthermore, applying CycleDiffusion to text-to-image diffusion models, we show that large-scale text-to-image diffusion models can be used as **zero-shot** image-to-image editors. (2) One can guide pretrained diffusion models and GANs by controlling the latent codes in a unified, plug-and-play formulation based on energy-based models. Using the CLIP model and a face recognition model as guidance, we demonstrate that diffusion models have better coverage of low-density sub-populations and individuals than GANs.[1]

## 1 Introduction

Diffusion models (Song & Ermon, 2019; Ho et al., 2020) have achieved unprecedented results in generative modeling and are instrumental to text-to-image models such as DALL·E 2 (Ramesh et al., 2022). Unlike GANs (Goodfellow et al., 2014), VAEs (Kingma & Welling, 2014), and normalizing flows (Dinh et al., 2015), which have a simple (e.g., Gaussian) latent space, the commonly-adopted formulation of the "latent code" of diffusion models is a sequence of gradually denoised images. This formulation makes the prior distribution of the "latent code" data-dependent, deviating from the idea that generative models are mappings from simple noises to data (Goodfellow et al., 2014).

This paper provides a unified view of generative models of images by reformulating various diffusion models as deterministic maps from a Gaussian latent code $z$ to an image $x$ (Figure 1, Section 3.1). A question that follows is *encoding*: how to map an image $x$ to a latent code $z$. Encoding has been studied for many generative models. For instance, VAEs and normalizing flows have encoders by design, GAN inversion (Xia et al., 2021) builds *post hoc* encoders for GANs, and deterministic diffusion probabilistic models (DPMs) (Song et al., 2021a;b) build encoders with forward ODEs. However, it is still unclear how to build an encoder for stochastic DPMs such as DDPM (Ho et al., 2020), non-deterministic DDIM (Song et al., 2021a), and latent diffusion models (Rombach et al., 2022). We propose **DPM-Encoder** (Section 3.2), a reconstructable encoder for stochastic DPMs.

We show that some intriguing consequences emerge from our definition of the latent space of diffusion models and our DPM-Encoder. First, observations have been made that, given two diffusion models, a fixed "random seed" produces similar images (Nichol et al., 2022). Under our formulation, we formalize "similar images" via an upper bound of image distances. Since the defined latent code contains all randomness during sampling, DPM-Encoder is similar-in-spirit to inferring the "random seed" from real images. Based on this intuition and the upper bound of image distances, we propose **CycleDiffusion** (Section 3.3), a method for unpaired image-to-image translation using

---

[1]Our codes will be publicly available.

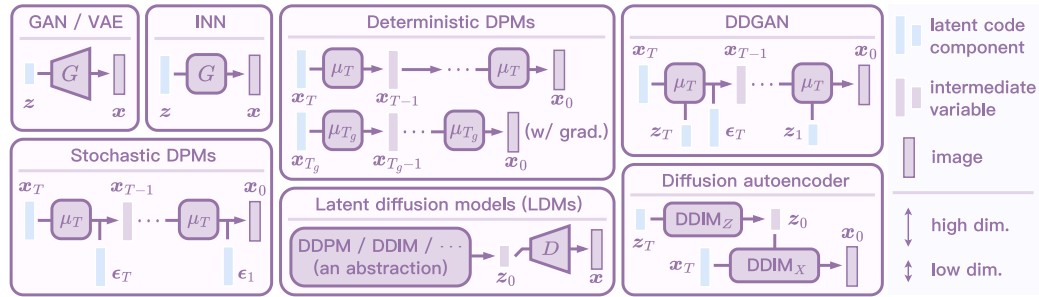

Figure 1: Once trained, various types of diffusion models can be reformulated as deterministic maps from latent code $z$ to image $x$, like GANs, VAEs, and normalizing flows.

our DPM-Encoder. Like the GAN-based UNIT method (Liu et al., 2017), CycleDiffusion encodes and decodes images using the common latent space. Our experiments show that CycleDiffusion outperforms previous methods based on GANs or diffusion models (Section 4.1). Furthermore, by applying large-scale text-to-image diffusion models (e.g., Stable Diffusion; Rombach et al., 2022) to CycleDiffusion, we obtain **zero-shot** image-to-image editors (Section 4.2).

With a simple latent prior, generative models can be guided in a plug-and-play manner by means of energy-based models (Nguyen et al., 2017; Nie et al., 2021; Wu et al., 2022). Thus, our unification allows unified, plug-and-play guidance for various diffusion models and GANs (Section 3.4), which avoids finetuning the guidance model on noisy images for diffusion models (Dhariwal & Nichol, 2021; Liu et al., 2021). With the CLIP model and a face recognition model as guidance, we show that diffusion models have broader coverage of low-density sub-populations and individuals (Section 4.3).

## 2 RELATED WORK

Recent years have witnessed a great progress in generative models, such as GANs (Goodfellow et al., 2014), diffusion models (Song & Ermon, 2019; Ho et al., 2020; Dhariwal & Nichol, 2021), VAEs (Kingma & Welling, 2014), normalizing flows (Dinh et al., 2015), and their hybrid extensions (Sinha et al., 2021; Vahdat et al., 2021; Zhang & Chen, 2021; Kim et al., 2022a). Previous works have shown that their training objectives are related, e.g., diffusion models as VAEs (Ho et al., 2020; Kingma et al., 2021; Huang et al., 2021); GANs and VAEs as KL divergences (Hu et al., 2018) or mutual information with consistency constraints (Zhao et al., 2018); a recent attempt (Zhang et al., 2022b) has been made to unify several generative models as GFlowNets (Bengio et al., 2021). In contrast, this paper unifies generative models as deterministic mappings from Gaussian noises to data (*aka* implicit models) once they are trained. Generative models with non-Gaussian randomness (Davidson et al., 2018; Nachmani et al., 2021) can be unified as deterministic mappings in similar ways.

One of the most fundamental challenges in generative modeling is to design an encoder that is both computationally efficient and invertible. GAN inversion trains an encoder after GANs are pre-trained (Xia et al., 2021). VAEs and normalizing flows have their encoders by design. Song et al. (2021a;b) studied encoding for ODE-based deterministic diffusion probabilistic models (DPMs). However, it remains unclear how to encode for general stochastic DPMs, and DPM-Encoder fills this gap. Also, CycleDiffusion can be seen as an extension of Su et al. (2022)'s DDIB approach to stochastic DPMs.

Previous works have formulated plug-and-play guidance of generative models as latent-space energy-based models (EBMs) (Nguyen et al., 2017; Nie et al., 2021; Wu et al., 2022), and our unification makes it applicable to various diffusion models, which are effective for modeling images, audio (Kong et al., 2021), videos (Ho et al., 2022; Hoppe et al., 2022), molecules (Xu et al., 2022), 3D objects (Luo & Hu, 2021), and text (Li et al., 2022). This plug-and-play guidance can provide principled, fine-grained model comparisons of coverage of sub-populations and individuals on the same dataset.

A concurrent work observed that fixing both (1) the random seed and (2) the cross-attention map in Transformer-based text-to-image diffusion models results in images with minimal changes (Hertz et al., 2022). The idea of fixing the cross-attention map is named Cross Attention Control (CAC) in that work, which can be used to edit *model-generated* images *when the random seed is known*.

For *real* images with stochastic DPMs, they generate masks based on the attention map because the random seed is unknown for real images. In Section 4.2, we show that CycleDiffusion and CAC can be combined to improve the structural preservation of image editing.

Table 1: Details of redefining various diffusion models' latent space (Section 3.1).

| | Latent code $z$ | Deterministic map $x = G(z)$ |
|---|---|---|
| Stochastic DPMs | $z := (x_T \oplus \epsilon_T \oplus \cdots \oplus \epsilon_1)$ | $x_{T-1} = \mu_T(x_T, T) + \sigma_T \odot \epsilon_T,$ $x_{t-1} = \mu_T(x_t, t) + \sigma_t \odot \epsilon_t \ (t < T), \quad x := x_0.$ |
| Deterministic DPMs | $z := x_T$ ($T = T_g$ if with gradient) | $x_{T-1} = \mu_T(x_T, T),$ $x_{t-1} = \mu_T(x_t, t) \ (t < T), \quad x := x_0.$ |
| LDM | $z$ of $G_{\text{latent}}$ | $z_0 = G_{\text{latent}}(z), \quad x = D(z_0).$ |
| DiffAE | $z := (z_T \oplus x_T)$ | $z_0 = \text{DDIM}_Z(z_T), \quad x := x_0 = \text{DDIM}_X(x_T, z_0).$ |
| DDGAN | $z := (x_T \oplus z_T \oplus \epsilon_T \oplus \cdots$ $\oplus z_2 \oplus \epsilon_2 \oplus z_1)$ | $x_{T-1} = \mu_T(x_T, z_T, T) + \sigma_T \odot \epsilon_T,$ $x_{t-1} = \mu_T(x_t, z_t, t) + \sigma_t \odot \epsilon_t \ (1 < t < T),$ $x := x_0 = \mu_T(x_1, z_1, 1).$ |

# 3 METHOD

## 3.1 GAUSSIAN LATENT SPACE FOR DIFFUSION MODELS

Generative models such as GANs, VAEs, and normalizing flows can be seen as a family of *implicit models*, meaning that they are deterministic maps $G : \mathbb{R}^d \to \mathcal{X}$ from latent codes $z$ to images $x$. At inference, sampling from the image prior $x \sim p_x(x)$ is implicitly defined as $z \sim p_z(z), x = G(z)$. The latent prior $p_z(z)$ is commonly chosen to be the isometric Gaussian distribution. In this section, we show how to unify diffusion models into this family. Overview is shown in Figure 1 and Table 1.

**Stochastic DPMs:** Stochastic DPMs (Ho et al., 2020; Song & Ermon, 2019; Song et al., 2021b;a; Watson et al., 2022) generate images with a Markov chain. Given the mean estimator $\mu_T$ (see Appendix A) and $x_T \sim \mathcal{N}(\mathbf{0}, \mathbf{I})$, the image $x := x_0$ is generated through $x_{t-1} \sim \mathcal{N}(\mu_T(x_t, t), \text{diag}(\sigma_t^2))$. Using the reparameterization trick, we define the latent code $z$ and the mapping $G$ recursively as

$$z := (x_T \oplus \epsilon_T \oplus \cdots \oplus \epsilon_1) \sim \mathcal{N}(\mathbf{0}, \mathbf{I}), \quad x_{t-1} = \mu_T(x_t, t) + \sigma_t \odot \epsilon_t, \quad t = T, \ldots, 1, \quad (1)$$

where $\oplus$ is concatenation. Here, $z$ has dimension $d = d_I \times (T+1)$, where $d_I$ is the image dimension.

**Deterministic DPMs:** Deterministic DPMs (Song et al., 2021a;b; Salimans & Ho, 2022; Liu et al., 2022; Lu et al., 2022; Karras et al., 2022; Zhang & Chen, 2022) generate images with the ODE formulation. Given the mean estimator $\mu_T$, deterministic DPMs generate $x := x_0$ via

$$z := x_T \sim \mathcal{N}(\mathbf{0}, \mathbf{I}), \quad x_{t-1} = \mu_T(x_t, t), \quad t = T, \ldots, 1. \quad (2)$$

Since backpropagation through Eq. (2) is costly, we use fewer discretization steps $T_g$ when computing gradients. Given the mean estimator $\mu_{T_g}$ with number of steps $T_g$, the image $x := x_0$ is generated as

$$\text{(with gradients)} \quad z := x_{T_g} \sim \mathcal{N}(\mathbf{0}, \mathbf{I}), \quad x_{t-1} = \mu_{T_g}(x_t, t), \quad t = T_g, \ldots, 1. \quad (3)$$

**Latent diffusion models (LDMs):** An LDM (Rombach et al., 2022) first uses a diffusion model $G_{\text{latent}}$ to compute a "latent code" $z_0 = G_{\text{latent}}(z)$,[2] which is then decoded as $x = D(z_0)$. Note that $G_{\text{latent}}$ is an abstraction of the diffusion models that are already unified above.

**Diffusion autoencoder (DiffAE):** DiffAE (Preechakul et al., 2022) first uses a deterministic DDIM to generate a "latent code" $z_0$,[2] which is used as condition for an image-space deterministic DDIM:

$$z := (z_T \oplus x_T) \sim \mathcal{N}(\mathbf{0}, \mathbf{I}), \quad z_0 = \text{DDIM}_Z(z_T), \quad x := x_0 = \text{DDIM}_X(x_T, z_0). \quad (4)$$

**DDGAN:** DDGAN (Xiao et al., 2022) models each reverse time step $t$ as a GAN conditioned on the output of the previous step. We define the latent code $z$ and generation process $G$ of DDGAN as

$$z := (x_T \oplus z_T \oplus \epsilon_T \oplus \cdots \oplus z_2 \oplus \epsilon_2 \oplus z_1) \sim \mathcal{N}(\mathbf{0}, \mathbf{I}),$$
$$x_{t-1} = \mu_T(x_t, z_t, t) + \sigma_t \odot \epsilon_t, \quad t = T, \ldots, 2, \quad x := x_0 = \mu_T(x_1, z_1, 1). \quad (5)$$

---

[2]Quotation marks stand for "latent code" in the cited papers, different from our latent code $z$ in Section 3.1.

---

**Algorithm 1:** CycleDiffusion for zero-shot image-to-image translation

---

**Input:** source image $\boldsymbol{x} := \boldsymbol{x}_0$; source text $\boldsymbol{t}$; target text $\hat{\boldsymbol{t}}$; encoding step $T_{\text{es}} \leq T$

1. Sample noisy image $\hat{\boldsymbol{x}}_{T_{\text{es}}} = \boldsymbol{x}_{T_{\text{es}}} \sim q(\boldsymbol{x}_{T_{\text{es}}}|\boldsymbol{x}_0)$

**for** $t = T_{es}, \ldots, 1$ **do**

    2. $\boldsymbol{x}_{t-1} \sim q(\boldsymbol{x}_{t-1}|\boldsymbol{x}_t, \boldsymbol{x}_0)$

    3. $\boldsymbol{\epsilon}_t = \big(\boldsymbol{x}_{t-1} - \boldsymbol{\mu}_T(\boldsymbol{x}_t, t|\boldsymbol{t})\big)/\boldsymbol{\sigma}_t$

    4. $\hat{\boldsymbol{x}}_{t-1} = \boldsymbol{\mu}_T(\hat{\boldsymbol{x}}_t, t|\hat{\boldsymbol{t}}) + \boldsymbol{\sigma}_t \odot \boldsymbol{\epsilon}_t$

**Output:** $\hat{\boldsymbol{x}} := \hat{\boldsymbol{x}}_0$

---

## 3.2 DPM-ENCODER: A RECONSTRUCTABLE ENCODER FOR DIFFUSION MODELS

In this section, we investigate the *encoding* problem, i.e., $\boldsymbol{z} \sim \text{Enc}(\boldsymbol{z}|\boldsymbol{x}, G)$. The encoding problem has been studied for many generative models, and our contribution is **DPM-Encoder**, an encoder for stochastic DPMs. DPM-Encoder is defined as follows. For each image $\boldsymbol{x} := \boldsymbol{x}_0$, stochastic DPMs define a posterior distribution $q(\boldsymbol{x}_{1:T}|\boldsymbol{x}_0)$ (Ho et al., 2020; Song et al., 2021a). Based on $q(\boldsymbol{x}_{1:T}|\boldsymbol{x}_0)$ and Eq. (1), we can directly derive $\boldsymbol{z} \sim \text{DPMEnc}(\boldsymbol{z}|\boldsymbol{x}, G)$ as (see details in Appendices A and B)

$$\boldsymbol{x}_1, \ldots, \boldsymbol{x}_{T-1}, \boldsymbol{x}_T \sim q(\boldsymbol{x}_{1:T}|\boldsymbol{x}_0), \quad \boldsymbol{\epsilon}_t = \big(\boldsymbol{x}_{t-1} - \boldsymbol{\mu}_T(\boldsymbol{x}_t, t)\big)/\boldsymbol{\sigma}_t, \quad t = T, \ldots, 1,$$
$$\boldsymbol{z} := \big(\boldsymbol{x}_T \oplus \boldsymbol{\epsilon}_T \oplus \cdots \oplus \boldsymbol{\epsilon}_2 \oplus \boldsymbol{\epsilon}_1\big). \tag{6}$$

A property of DPM-Encoder is perfect reconstruction, meaning that we have $\boldsymbol{x} = G(\boldsymbol{z})$ for every $\boldsymbol{z} \sim \text{Enc}(\boldsymbol{z}|\boldsymbol{x}, G)$. A proof by induction is provided in Appendix B.

## 3.3 CYCLEDIFFUSION: IMAGE-TO-IMAGE TRANSLATION WITH DPM-ENCODER

Given two stochastic DPMs $G_1$ and $G_2$ that model two distributions $\mathcal{D}_1$ and $\mathcal{D}_2$, several researchers and practitioners have found that sampling with the same "random seed" leads to similar images (Nichol et al., 2022). To formalize "similar images", we provide an upper bound of image distances based on assumptions about the trained DPMs, shown at the end of this subsection. Based on this finding, we propose a simple unpaired image-to-image translation method, CycleDiffusion. Given a source image $\boldsymbol{x} \in \mathcal{D}_1$, we use DPM-Encoder to encode it as $\boldsymbol{z}$ and then decode it as $\hat{\boldsymbol{x}} = G_2(\boldsymbol{z})$:

$$\boldsymbol{z} \sim \text{DPMEnc}(\boldsymbol{z}|\boldsymbol{x}, G_1), \quad \hat{\boldsymbol{x}} = G_2(\boldsymbol{z}). \tag{7}$$

We can also apply CycleDiffusion to text-to-image diffusion models by defining $\mathcal{D}_1$ and $\mathcal{D}_2$ as image distributions conditioned on two texts. Let $G_{\boldsymbol{t}}$ be a text-to-image diffusion model conditioned on text $\boldsymbol{t}$. Given a source image $\boldsymbol{x}$, the user writes two texts: a source text $\boldsymbol{t}$ describing the source image $\boldsymbol{x}$ and a target text $\hat{\boldsymbol{t}}$ describing the target image $\hat{\boldsymbol{x}}$ to be generated. We can then perform zero-shot image-to-image editing via (zero-shot means that the model has never been trained on image editing)

$$\boldsymbol{z} \sim \text{DPMEnc}(\boldsymbol{z}|\boldsymbol{x}, G_{\boldsymbol{t}}), \quad \hat{\boldsymbol{x}} = G_{\hat{\boldsymbol{t}}}(\boldsymbol{z}). \tag{8}$$

Inspired by the realism-faithfulness tradeoff in SDEdit (Meng et al., 2022), we can truncate $\boldsymbol{z}$ towards a specified encoding step $T_{\text{es}} \leq T$. The algorithm of CycleDiffusion is shown in Algorithm 1.

**Why does a fixed $\boldsymbol{z}$ leads to similar images?** We illustrate with text-to-image diffusion models. Suppose the text-to-image model has the following two properties:

1. Conditioned on the same text, similar noisy images lead to similar enough mean predictions. Formally, $\boldsymbol{\mu}_T(\boldsymbol{x}_t, t|\boldsymbol{t})$ is $K_t$-Lipschitz, i.e., $\|\boldsymbol{\mu}_T(\boldsymbol{x}_t, t|\boldsymbol{t}) - \boldsymbol{\mu}_T(\hat{\boldsymbol{x}}_t, t|\boldsymbol{t})\| \leq K_t\|\boldsymbol{x}_t - \hat{\boldsymbol{x}}_t\|$.

2. Given the same image, the two texts lead to similar predictions. Formally, $\|\boldsymbol{\mu}_T(\boldsymbol{x}_t, t|\boldsymbol{t}) - \boldsymbol{\mu}_T(\boldsymbol{x}_t, t|\hat{\boldsymbol{t}})\| \leq S_t$. Intuitively, a smaller difference between $\boldsymbol{t}$ and $\hat{\boldsymbol{t}}$ gives us a smaller $S_t$.

Let $B_t$ be the upper bound of $\|\boldsymbol{x}_t - \hat{\boldsymbol{x}}_t\|_2$ at time step $t$ when the same latent code $\boldsymbol{z}$ is used for sampling (i.e., $\boldsymbol{x}_0 = G_{\boldsymbol{t}}(\boldsymbol{z})$ and $\hat{\boldsymbol{x}}_0 = G_{\hat{\boldsymbol{t}}}(\boldsymbol{z})$). We have $B_T^s = 0$ because $\|\boldsymbol{x}_T - \hat{\boldsymbol{x}}_t\|_2 = 0$, and $B_0^s$ is the upper bound for the generated images $\|\boldsymbol{x} - \hat{\boldsymbol{x}}\|_2$. The upper bound $B_t$ can be propagated through time, from $T$ to 0. Specifically, by combining the above two properties, we have

$$B_{t-1} \leq (K_t + 1)B_t + S_t \tag{9}$$

## 3.4 Unified Plug-and-Play Guidance for Generative Models

Prior works showed that guidance for generative models can be achieved in the latent space (Nguyen et al., 2017; Nie et al., 2021; Wu et al., 2022). Specifically, given a condition $\mathcal{C}$, one can define the guided image distribution as an energy-based model (EBM): $p(\boldsymbol{x}|\mathcal{C}) \propto p_{\boldsymbol{x}}(\boldsymbol{x})e^{-\lambda_{\mathcal{C}}E(\boldsymbol{x}|\mathcal{C})}$. Sampling for $\boldsymbol{x} \sim p(\boldsymbol{x}|\mathcal{C})$ is equivalent to $\boldsymbol{z} \sim p_{\boldsymbol{z}}(\boldsymbol{z}|\mathcal{C}), \boldsymbol{x} = G(\boldsymbol{z})$, where $p(\boldsymbol{z}|\mathcal{C}) \propto p_{\boldsymbol{z}}(\boldsymbol{z})e^{-\lambda_{\mathcal{C}}E(G(\boldsymbol{z})|\mathcal{C})}$. Examples of the energy function $E(\boldsymbol{x}|\mathcal{C})$ are provided in Section 4.3. To sample $\boldsymbol{z} \sim p(\boldsymbol{z}|\mathcal{C})$, one can use any model-agnostic samplers. For example, Langevin dynamics (Welling & Teh, 2011) starts from $\boldsymbol{z}^{\langle 0 \rangle} \sim \mathcal{N}(\mathbf{0}, \boldsymbol{I})$ and samples $\boldsymbol{z} := \boldsymbol{z}^{\langle n \rangle}$ iteratively through

$$\boldsymbol{z}^{\langle k+1 \rangle} = \boldsymbol{z}^{\langle k \rangle} + \frac{\sigma}{2}\nabla_{\boldsymbol{z}}\Big(\log p_{\boldsymbol{z}}(\boldsymbol{z}^{\langle k \rangle}) - E\big(G(\boldsymbol{z}^{\langle k \rangle})|\mathcal{C}\big)\Big) + \sqrt{\sigma}\boldsymbol{\omega}^{\langle k \rangle}, \quad \boldsymbol{\omega}^{\langle k \rangle} \sim \mathcal{N}(\mathbf{0}, \boldsymbol{I}). \quad (10)$$

Table 2: Quantitative comparison for unpaired image-to-image translation methods. Methods in the second block use the same pre-trained diffusion model in the target domain. Results of CUT, ILVR, SDEdit, and EGSDE are from Zhao et al. (2022). Best results using diffusion models are in **bold**. CycleDiffusion has the best FID and KID among all methods and the best SSIM among methods with diffusion models. Note that it has been shown that SSIM is much better correlated with human visual perception than squared distance-based metrics such as $L_2$ and PSNR (Wang et al., 2004).

| | Cat → Dog | | | | Wild → Dog | | | |
|---|---|---|---|---|---|---|---|---|
| | FID↓ | KID×$10^3$↓ | PSNR↑ | SSIM↑ | FID↓ | KID×$10^3$↓ | PSNR↑ | SSIM↑ |
| CUT (GAN SOTA; Park et al., 2020) | 76.21 | – | 17.48 | 0.601 | 92.94 | – | 17.20 | 0.592 |
| ILVR (Choi et al., 2021) | 74.37 | – | 17.77 | 0.363 | 75.33 | – | 16.85 | 0.287 |
| SDEdit (Meng et al., 2022) | 74.17 | – | 19.19 | 0.423 | 68.51 | – | 17.98 | 0.343 |
| EGSDE (Zhao et al., 2022) | 65.82 | – | **19.31** | 0.415 | 59.75 | – | **18.14** | 0.343 |
| CycleDiffusion w/ DDIM ($\eta = 0.1$) | **58.87** | 20.3 | 18.50 | **0.557** | **56.45** | 19.5 | 17.82 | **0.479** |

## 4 Experiments

This section provides experimental validation of the proposed work. Section 4.1 shows how CycleDiffusion achieves competitive results on unpaired image-to-image translation benchmarks. Section 4.2 provides a protocol for what we call zero-shot image-to-image translation; CycleDiffusion outperforms several image-to-image translation baselines that we re-purposed for this new task. Section 4.3 shows how diffusion models and GANs can be guided in a unified, plug-and-play formulation.

### 4.1 CycleDiffusion for Unpaired Image-to-Image Translation

Given two unaligned image domains, unpaired image-to-image translation aims at mapping images in one domain to the other. We follow setups from previous works whenever possible, as detailed below. Following previous work (Park et al., 2020; Zhao et al., 2022), we conducted experiments on the test set of AFHQ (Choi et al., 2020) with resolution $256 \times 256$ for Cat → Dog and Wild → Dog. For each source image, each method should generate a target image with minimal changes. Since CycleDiffusion sometimes generates noisy outputs, we used $T_{\text{sdedit}}$ steps of SDEdit for denoising. When $T = 1000$, we set $T_{\text{sdedit}} = 100$ for Cat → Dog and $T_{\text{sdedit}} = 125$ for Wild → Dog.

**Metrics:** To evaluate realism, we reported Frechet Inception Distance (FID; Heusel et al., 2017) and Kernel Inception Distance (KID; Bińkowski et al., 2018) between the generated and target images. To evaluate faithfulness, we reported Peak Signal-to-Noise Ratio (PSNR) and Structural Similarity Index Measure (SSIM; Wang et al., 2004) between each generated image and its source image.

**Baselines:** We compared CycleDiffusion with previous state-of-the-art unpaired image-to-image translation methods: CUT (Park et al., 2020), ILVR (Choi et al., 2021), SDEdit (Meng et al., 2022), and EGSDE (Zhao et al., 2022). CUT is based on GAN, and the others use diffusion models.

**Pre-trained diffusion models:** ILVR, SDEdit, and EGSDE only need the diffusion model trained on the target domain, and we followed them to use the pre-trained model from Choi et al. (2021) for Dog. CycleDiffusion needs diffusion models on both domains, so we trained them on Cat and Wild.

Seen in Table 2 are the results. CycleDiffusion has the best realism (i.e., FID and KID). There is a mismatch between the faithfulness metrics (i.e., PSNR and SSIM), and note that SSIM is much better correlated with human perception than PSNR (Wang et al., 2004). Among all diffusion model-based methods, CycleDiffusion achieves the highest SSIM. Figure 2 displays some image samples from CycleDiffusion, showing that our method can change the domain while preserving local details such as the background, lighting, pose, and overall color the animal.

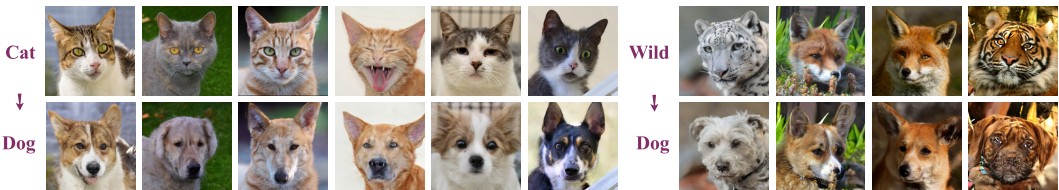

Figure 2: Unpaired image-to-image translation (Cat → Dog, Wild → Dog) with CycleDiffusion.

## 4.2 TEXT-TO-IMAGE DIFFUSION MODELS CAN BE ZERO-SHOT IMAGE-TO-IMAGE EDITORS

This section provides experiments for zero-shot image editing. We curated a set of 150 tuples $(\boldsymbol{x}, \boldsymbol{t}, \hat{\boldsymbol{t}})$ for this task, where $\boldsymbol{x}$ is the source image, $\boldsymbol{t}$ is the source text (e.g., "an aerial view of autumn scene." in Figure 3 second row on the right), and $\hat{\boldsymbol{t}}$ is the target text (e.g., "an aerial view of winter scene."). The generated image is denoted as $\hat{\boldsymbol{x}}$. We also demonstrate that CycleDiffusion can be combined with the Cross Attention Control (Hertz et al., 2022) to further preserve the image structure.

**Metrics:** To evaluate faithfulness to the source image, we reported PSNR and SSIM. To evaluate authenticity to the target text, we reported the CLIP score $\mathcal{S}_{\text{CLIP}}(\hat{\boldsymbol{x}}|\hat{\boldsymbol{t}}) = \cos\left\langle \text{CLIP}_{\text{img}}(\hat{\boldsymbol{x}}), \text{CLIP}_{\text{text}}(\hat{\boldsymbol{t}})\right\rangle$, where the CLIP embeddings are normalized. We note a trade-off between PSNR/SSIM and $\mathcal{S}_{\text{CLIP}}$: by copying the source image we get high PSNR/SSIM but low $\mathcal{S}_{\text{CLIP}}$, and by ignoring the source image (e.g., by directly generating images conditioned on the target text) we get high $\mathcal{S}_{\text{CLIP}}$ but low PSNR/SSIM. To address this trade-off, we also reported the directional CLIP score (Patashnik et al., 2021) (both CLIP embeddings and the embedding differences are normalized):

$$\mathcal{S}_{\text{D-CLIP}}(\hat{\boldsymbol{x}}|\boldsymbol{x}, \boldsymbol{t}, \hat{\boldsymbol{t}}) = \cos\left\langle \text{CLIP}_{\text{img}}(\hat{\boldsymbol{x}}) - \text{CLIP}_{\text{img}}(\boldsymbol{x}), \text{CLIP}_{\text{text}}(\hat{\boldsymbol{t}}) - \text{CLIP}_{\text{text}}(\boldsymbol{t})\right\rangle. \quad (11)$$

**Baselines:** The baselines include SDEdit (Meng et al., 2022) and DDIB (Su et al., 2022). We used the same hyperparameters for the baselines and CycleDiffusion whenever possible (e.g., the number of diffusion steps, the strength of classifier-free guidance; see Appendix C).

**Pre-trained text-to-image diffusion models:** To investigate how the zero-shot performance changes with data size, data quality, and training details, we used the following models: (1) `LDM-400M`, a 1.45B-parameter model trained on LAION-400M (Schuhmann et al., 2021), (2) `SD-v1-1`, a 0.98B-parameter Stable Diffusion model trained on LAION-5B (Schuhmann et al., 2022), (3) `SD-v1-4`, finetuned from `SD-v1-1` for improved aesthetics and classifier-free guidance.

**Results:** Table 3 shows the results for zero-shot image-to-image translation. CycleDiffusion excels at being faithful to the source image (i.e., PSNR and SSIM); by contrast, SDEdit and DDIB have comparable authenticity to the target text (i.e., $\mathcal{S}_{\text{CLIP}}$), but their outputs are much less faithful. For all methods, we find that the pre-trained weights `SD-v1-1` and `SD-v1-4` have better faithfulness than `LDM-400M`. Figure 3 provides samples from CycleDiffusion, demonstrating that CycleDiffusion achieves meaningful edits that span (1) replacing objects, (2) adding objects, (3) changing styles, and (4) modifying attributes. See Figure 7 (Appendix E) for qualitative comparisons with the baselines.

**CycleDiffusion + Cross Attention Control:** Besides fixing the random seed, Hertz et al. (2022) shows that fixing the cross attention map (i.e., Cross Attention Control, or CAC) further improves the similarity between synthesized images. CAC is applicable to CycleDiffusion: in Algorithm 1, we can apply the attention map of $\boldsymbol{\mu}_T(\boldsymbol{x}_t, t|\boldsymbol{t})$ to $\boldsymbol{\mu}_T(\boldsymbol{x}_t, t|\hat{\boldsymbol{t}})$. However, we cannot apply it to all samples because CAC puts requirements on the difference between $\boldsymbol{t}$ and $\hat{\boldsymbol{t}}$. Figure 4 shows that CAC helps CycleDiffusion when the intended *structural* change is small. For instance, when the intended change is color but not shape (left), CAC helps CycleDiffusion preserve the background; when the intended change is horse → elephant, CAC makes the generated elephant to look more like a horse in shape.

Table 3: Zero-shot image editing. We did not use fixed hyperparameters, and neither did we plot the trade-off curve. The reason is that every input can have its best hyperparameters and even random seed (such per-sample tuning for Stable Diffusion is quite common nowadays). Instead, **for each input**, we ran 15 random trials for each hyperparameter and report the one with the highest $\mathcal{S}_{\text{D-CLIP}}$.

| | Method | $\mathcal{S}_{\text{CLIP}}\uparrow$ | $\mathcal{S}_{\text{D-CLIP}}\uparrow$ | PSNR$\uparrow$ | SSIM$\uparrow$ |
|---|---|---|---|---|---|
| LDM-400M | SDEdit (Meng et al., 2022) | 0.332 | 0.264 | 13.68 | 0.390 |
| | DDIB (Su et al., 2022) | 0.324 | 0.195 | 15.82 | 0.544 |
| | CycleDiffusion w/ DDIM ($\eta = 0.1$; ours) | **0.333** | **0.275** | **18.72** | **0.625** |
| SD-v1-1 | SDEdit (Meng et al., 2022) | **0.339** | 0.248 | 15.50 | 0.498 |
| | DDIB (Su et al., 2022) | 0.324 | 0.189 | 18.58 | 0.658 |
| | CycleDiffusion w/ DDIM ($\eta = 0.1$; ours) | 0.331 | **0.262** | **21.98** | **0.731** |
| SD-v1-4 | SDEdit (Meng et al., 2022) | **0.344** | 0.258 | 15.93 | 0.512 |
| | DDIB (Su et al., 2022) | 0.331 | 0.209 | 18.10 | 0.653 |
| | CycleDiffusion w/ DDIM ($\eta = 0.1$; ours) | 0.334 | **0.272** | **21.92** | **0.731** |

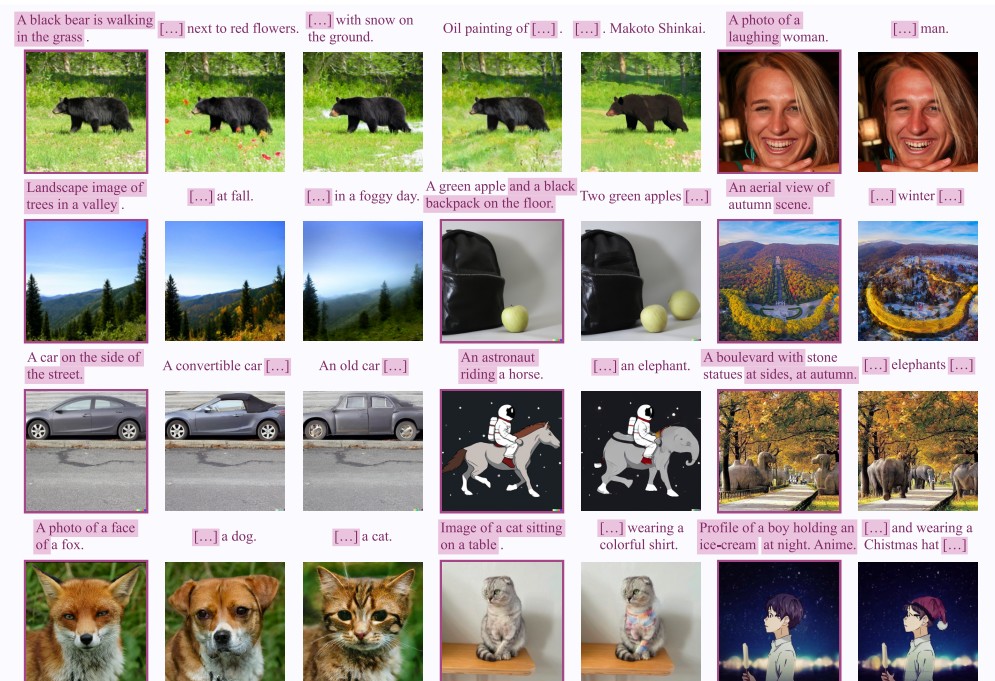

Figure 3: CycleDiffusion for zero-shot image editing. Source images $\boldsymbol{x}$ are displayed with a purple margin; the other images are the generated $\hat{\boldsymbol{x}}$. Within each pair of source and target texts, overlapping text spans are marked in purple in the source text and abbreviated as [. . .] in the target text.

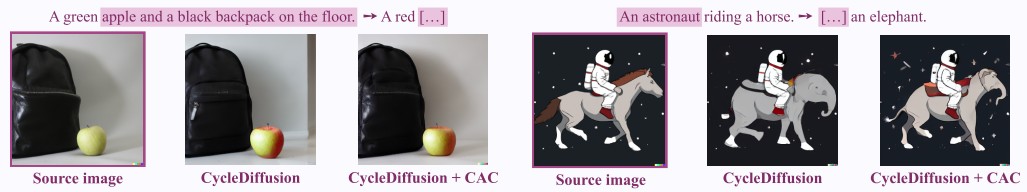

Figure 4: Cross Attention Control (CAC; Hertz et al., 2022) helps CycleDiffusion when the intended *structural* change is small. For instance, when the intended change is color but not shape (left), CAC helps CycleDiffusion preserve the background; when the intended change is horse $\rightarrow$ elephant, CAC makes the generated elephant look more like a horse in shape.

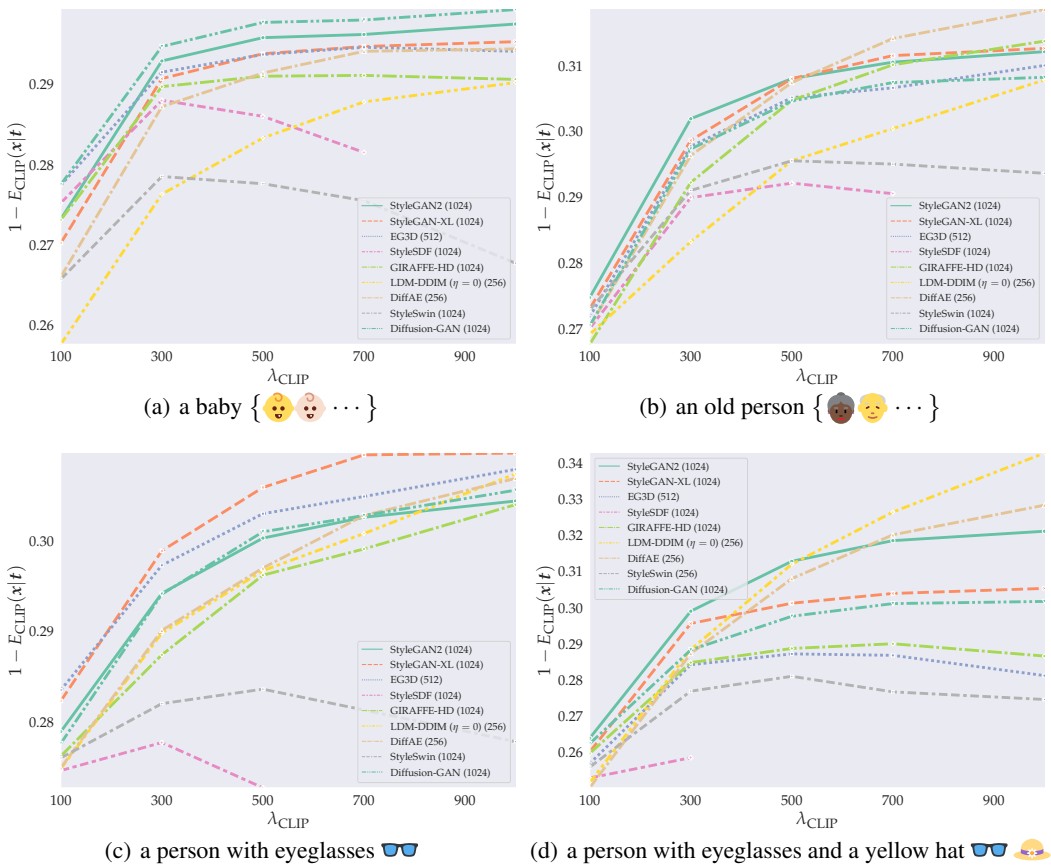

Figure 5: Unified plug-and-play guidance for diffusion models and GANs with text and CLIP. The text description used in each plot is a photo of $\boxed{\phantom{xxx}}$. Image samples and more analyses are in Figure 6 and Appendix G. When the guidance becomes complex, diffusion models surpass GANs.

## 4.3 UNIFIED PLUG-AND-PLAY GUIDANCE FOR DIFFUSION MODELS AND GANS

Previous methods for conditional sampling from (*aka* guiding) diffusion models require training the guidance model on noisy images (Dhariwal & Nichol, 2021; Liu et al., 2021), which deviates from the idea of plug-and-play guidance by leveraging the simple latent prior of generative models (Nguyen et al., 2017). In contrast, our definition of the Gaussian latent space of different diffusion models allows for unified plug-and-play guidance of diffusion models and GANs. It facilitates principled comparisons over sub-populations and individuals when models are trained on the same dataset.

We used the text $t$ to specify sub-population. For instance, a photo of baby represents the baby sub-population in the domain of human faces. We instantiate the energy in Section 3.4 as $E_{\text{CLIP}}(x|t) = \frac{1}{L}\sum_{l=1}^{L}\left(1 - \cos\left\langle\text{CLIP}_{\text{img}}\left(\text{DiffAug}_l(x)\right), \text{CLIP}_{\text{text}}(t)\right\rangle\right)$, where $\text{DiffAug}_l$ stands for differentiable augmentation (Zhao et al., 2020) that mitigates the adversarial effect, and we sample from the energy-based distribution using Langevin dynamics in Eq. (10) with $n = 200$, $\sigma = 0.05$. We enumerated the guidance strength (i.e., the coefficient $\lambda_{\mathcal{C}}$ in Section 3.4) $\lambda_{\text{CLIP}} \in \{100, 300, 500, 700, 1000\}$. For evaluation, we reported $(1 - E_{\text{CLIP}}(x|t))$ averaged over 256 samples. This metric quantifies whether the sampled images are consistent with the specified text $t$. Figure 5 plots models with pre-trained weights on FFHQ (Karras et al., 2019) (citations in Table 5, Appendix G). In Figure 6, we visualize samples for SN-DDPM and DDGAN trained on CelebA. We find that diffusion models outperform 2D/3D GANs for complex text, and different models represent the same sub-population differently.

Broad coverage of individuals is an important aspect of the personalized use of generative models. To analyze this coverage, we guide different models to generate images that are close to a reference

$\boldsymbol{x}_r$ in the identity (ID) space modeled by the IR-SE50 face embedding model (Deng et al., 2019), denoted as $R$. Given an ID reference image $\boldsymbol{x}_r$, we instantiated the energy defined in Section 3.4 as $E_{\text{ID}}(\boldsymbol{x}|\boldsymbol{x}_r) = 1 - \cos\langle R(\boldsymbol{x}), R(\boldsymbol{x}_r)\rangle$ with strength $\lambda_{\text{ID}} = 2500$ (i.e., $\lambda_{\mathcal{C}}$ in Section 3.4). For sampling, we used Langevin dynamics detailed in Eq. (10) with $n = 200$ and $\sigma = 0.05$. To measure ID similarity to the reference image $\boldsymbol{x}_r$, we reported $\cos\langle R(\boldsymbol{x}), R(\boldsymbol{x}_r)\rangle$, averaged over 256 samples. In Table 4, we report the performance of StyleGAN2, StyleGAN-XL, GIRAFFE-HD, EG3D, LDM-DDIM, DDGAN, and DiffAE. DDGAN is trained on CelebAHQ, while others are trained on FFHQ. We find that diffusion models have much better coverage of individuals than 2D/3D GANs. Among diffusion models, deterministic LDM-DDIM ($\eta = 0$) achieves the best identity guidance performance. We provide image samples of identity guidance in Figure 9 (Appendix G).

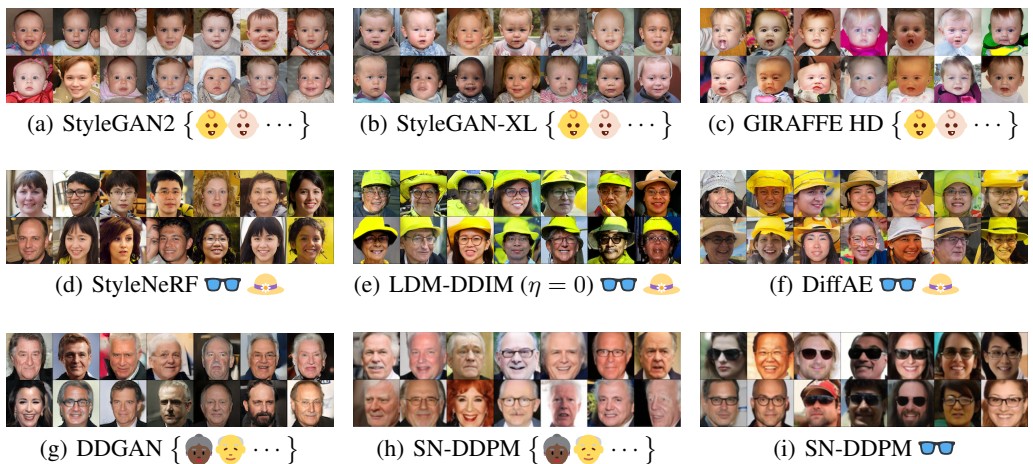

Figure 6: Sampling sub-populations from pre-trained generative models. Notations follow Figure 5.

Table 4: Guiding diffusion models and GANs with ID. ID-{A, B, C, D} are images from FFHQ. The metric is the ArcFace cosine similarity (Deng et al., 2019). See samples in Figure 9 (Appendix G).

|  | *2D GAN* | | *3D GAN* | | *Diffusion model* | | |
|---|---|---|---|---|---|---|---|
|  | StyleGAN2 | StyleGAN-XL | GIRAFFE-HD | EG3D | LDM-DDIM ($\eta = 0$) | DDGAN | DiffAE |
| ID-A | 0.561 | 0.681 | 0.616 | 0.468 | **0.904** | 0.837 | 0.873 |
| ID-B | 0.604 | 0.688 | 0.590 | 0.454 | **0.896** | 0.805 | 0.838 |
| ID-C | 0.495 | 0.636 | 0.457 | 0.403 | **0.892** | 0.795 | 0.852 |
| ID-D | 0.554 | 0.687 | 0.574 | 0.436 | **0.911** | 0.831 | 0.873 |

## 5 CONCLUSIONS AND DISCUSSION

This paper provides a unified view of pre-trained generative models by reformulating the latent space of diffusion models. While this reformulation is purely definitional, we show that it allows us to use diffusion models in similar ways as CycleGANs (Zhu et al., 2017) and GANs. Our CycleDiffusion achieves impressive performance on unpaired image-to-image translation (with two diffusion models trained on two domains independently) and zero-shot image-to-image translation (with text-to-image diffusion models). Our definition of latent code also allows diffusion models to be guided in the same way as GANs (i.e., plug-and-play, without finetuning on noisy images), and results show that diffusion models have broader coverage of sub-populations and individuals than GANs.

Besides the interesting results, it is worth noting that this paper raised more questions than provided answers. We have provided a formal analysis of the common latent space of stochastic DPMs via the bounded distance between images (Section 3.3), but it still needs further study. Notably, Khrulkov & Oseledets (2022) and Su et al. (2022) studied deterministic DPMs based on optimal transport. Furthermore, *efficient* plug-and-play guidance for stochastic DPMs on high-resolution images with many diffusion steps still remains open. These topics can be further explored in future studies.

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
