# OpenReview forum: "Unifying Diffusion Models' Latent Space, with Applications to CycleDiffusion and Guidance"
_ICLR.cc/2023/Conference — Submitted to ICLR 2023_

### Official Review · Reviewer_fAbM · 2022-10-27

**Confidence:** 4
**Clarity, Quality, Novelty And Reproducibility:** See above section.
**Correctness:** 3
**Technical Novelty And Significance:** 2
**Empirical Novelty And Significance:** 2
**Recommendation:** 3

**Strength And Weaknesses:**

Strength:
(1) The paper provides a clear summary for previous works.
(2) The paper is easy to digest.


Weaknesses:
(1) Limited Novelty: DPM-Encoder, which stores all the noises along the generative process, is an obvious way to encode an image to the latent space of a stochastic DPM. Since it is obvious, it appears in the famous work [1], Appendix B.2, to interpolate between images. Therefore, the encoding strategy is not new to me.
(2) Limited Novelty: CycleDiffusion is a simple generalization of SDEdit [2] and the observation in [3]. The authors made no additional theoretical justification on the method, making its improvement over previous works limited.
(3) Unsatisfying results: In Figure 3, many of the parts in the images, which we do not expect to change, changed. For example,  the astronaut figure on the bottom-left changes not only in the style, but also the content. This, in my opinion, is not fixable with the current method.

[1] Improved Techniques for Training Score-Based Generative Models, https://arxiv.org/pdf/2006.09011.pdf
[2] SDEdit: Guided Image Synthesis and Editing with Stochastic Differential Equations https://arxiv.org/abs/2108.01073
[3] Understanding DDPM Latent Codes Through Optimal Transport https://arxiv.org/abs/2202.07477


**Summary Of The Paper:**

The paper proposes to store the random noises along the generative trajectory of diffusion probabilistic model as a latent code, and use this latent code for image-to-image translation.

**Summary Of The Review:**

The paper leverages an existing strategy to encode images generated by DPM, then directly apply this latent code to another DPM to achieve image-to-image translation. The novelty is limited and the results are not satisfying.

---

> ### Author Response · Authors · 2022-11-14
> **Author response**
>
> > Limited Novelty: DPM-Encoder, which stores all the noises along the generative process, is an obvious way to encode an image to the latent space of a stochastic DPM. Since it is obvious, it appears in the famous work [1], Appendix B.2, to interpolate between images. Therefore, the encoding strategy is not new to me.
>
> -  We respectfully **disagree** with the claim “it [which refers to DPM-Encoder] appears in the famous work [1], Appendix B.2, to interpolate between images”. The interpolation in [1] is not between real images; therefore, **no encoding is needed in that experiment**. To make it more explicit, the interpolated $\boldsymbol{z}_1$ and $\boldsymbol{z}_2$ are two vectors sampled from the Gaussian prior instead of from the posterior that we define via DPM-Encoder. We hope that the reviewer could double-check this because it is important that we agree on facts.
>
> > Limited Novelty: CycleDiffusion is a simple generalization of SDEdit [2] and the observation in [3]. The authors made no additional theoretical justification for the method, making its improvement over previous works limited.
>
> - We would like to point out that [3] studies the deterministic ODE-based formulation (although it uses DDPM in the title). In the related work and experiment sections, we explicitly mentioned and compared with DDIB [4] (proposed in a concurrent work of [3], and we believe that it is exactly the same as the image translation method in [3]; we will also cite [3] when mentioning DDIB), and Table 4 shows that our method outperforms DDIB.
> - In the updated version, we provide a formal analysis of the *upper bound of image distances* to quantify “similar images”. The complete discussion can be found in the updated paper (at the end of page 4), but we would like to mention it here explicitly. Specifically, we can quantify two intuitions: (1) conditioned on the same text, similar noisy images lead to similar enough denoising directions; (2) Given the same image, two similar texts lead to similar predictions. Given (1) and (2), we show that fixing the latent code (or random seed) gives us a bound of image distance, which does not hold when not fixing the latent code (or random seed).
>
>
> > The paper leverages an existing strategy to encode images generated by DPM, then directly apply this latent code to another DPM to achieve image-to-image translation. The novelty is limited and the results are not satisfying.
>
> We respectfully **disagree** with the claim that DPM-Encoder is an existing strategy. An elaboration on why this is a piece of misunderstanding is detailed in our response to a previous question. Finally, we like the characterization of our proposed method as simple since we favor such straightforward solutions that work. We even pointed out in our paper that this formulation of latent code is by definition (in other words, simple). However, we would like to clarify two main contributions of this paper.
>
> 1. Our first contribution is DPM-Encoder, which makes it explicit how to sample latent codes from real images (i.e., a posterior distribution of latent codes) using stochastic diffusion models. For us, this is also simple or trivial, but it is very useful (as we show later) and to the best of our knowledge no one has proposed it before.
> 2. Our second contribution is to suggest two interesting applications: CycleDiffusion and plug-and-play guidance. More updated contributions such as a more formal analysis of the fact that fixed random noises lead to similar images, and the combination of CycleDiffusion and Cross Attention Control) are summarized below.
>
>
> **Updates of the paper**
>
> To avoid a verbatim copy of content, we refer you to our response to all reviewers, titled **Summary of updates**. Here, we briefly summarize the updates.
>
> 1. **CycleDiffusion + Cross Attention Control:** We showed that Cross Attention Control (i.e., fixing the cross-attention map) helps CycleDiffusion when the intended structural change is small.
> 2. **Formal analysis:** In the updated version, we quantify *similar images* as *bound between the distance the two images*. We show that fixing the latent code (or random seed) gives us a bound of image distance, which does not hold when not fixing the latent code (or random seed).  This analysis also suggests that more similar texts lead to more similar images when fixing the latent code.
> 3. **Code uploaded:** Our codes and instructions are now in the supplementary materials. After the review process, we will provide an implementation of CycleDiffusion with Stable Diffusion using the Huggingface diffusers library.
>
> We hope that you can consider the updates in the evaluation of this paper. We also look forward to your further comments and questions.
>
> [4] Dual Diffusion Implicit Bridges for Image-to-Image Translation. https://arxiv.org/abs/2203.08382

---

> > ### Comment · Reviewer_fAbM · 2022-11-16
> > **More Questions**
> >
> > 1. Thank you for the clarification. I agree that there are slight differences, but mathematically they share great similarity. In a high level, the encoding in [1] and this paper are both using the fact that 'same random seed generates same images in diffusion models', which is known to the field. Back to your specific argument that your method is encoding the image while the paper [1] is generating images, I would like to share another famous work [2]. In [2], Eq. (5) gives a simple SDE of how to encode an image into the noise space (your DPM-encoder space). Then, if you freeze the noise during running Eq. (5), like in [1], you get a latent code. I would argue this is equivalent to your DPM-encoder.
> >
> > [2] SCORE-BASED GENERATIVE MODELING THROUGH STOCHASTIC DIFFERENTIAL EQUATIONS, https://openreview.net/pdf?id=PxTIG12RRHS
> >
> > 2. The additional theory does not imply too much. It is just stating the fact that 'same random seed + same text results in the same image'. Actually, the landscape of diffusion models is complicated, and I'm not really sure if Lipschitz assumption holds.
> > 3. Reading your paper again, I have two more questions:
> >
> >     (1) In Eq. (10), how do you define $p_z(z)$?
> >
> >     (2) In Appendix F, how do you optimize $n$? As far as I know, $G(z)$ is a long chain and direct back-propagation is nearly impossible.

---

> > > ### Author Response · Authors · 2022-11-16
> > > **Trying to clarify a major misunderstanding**
> > >
> > > We believe that we know where we misunderstand each other: **our latent space is not the concatenation of noisy images; instead, our latent space is the concatenation of all isometric Gaussian noises in the denoising process**. Our responses to several questions below are based on clarifying this misunderstanding.
> > >
> > > > I agree that there are slight differences, but mathematically they share great similarity. In a high level, the encoding in [1] and this paper are both using the fact that 'same random seed generates same images in diffusion models', which is known to the field.
> > >
> > > - We would like to argue that the SDE formulation and ODE formulation are not slightly different. Specifically, the ODE formulation is *not* the limit of SDE when the variance term $g(t)$ goes to zero (please see the difference between Eq. (6) and Eq. (13) in the paper [2] you pointed out).
> > > - We don’t think there is a discussion on encoding in [1] (https://arxiv.org/pdf/2006.09011.pdf). However, we would like to further discuss if you could clarify it in a follow-up post. We would appreciate it!
> > >
> > >
> > > > Back to your specific argument that your method is encoding the image while the paper [1] is generating images, I would like to share another famous work [2]. In [2], Eq. (5) gives a simple SDE of how to encode an image into the noise space (your DPM-encoder space). [2] SCORE-BASED GENERATIVE MODELING THROUGH STOCHASTIC DIFFERENTIAL EQUATIONS
> > >
> > > - Thanks for double-checking [1], we are glad that we agree on the facts here.
> > > - However,  we respectfully **disagree** "Eq. (5) gives a simple SDE of how to encode an image into the noise space (your DPM-encoder space)". The output spaces of Eq. (5) in [2] and our DPM-encoder are different.
> > > - Eq. (5) in [2] is about **how to sample noisy images based on this clean image**. Following DDPM, we call this process the posterior distribution $q(\boldsymbol{x}_\{1:T}|\boldsymbol{x}_0)$ in our paper. This posterior distribution is a component of DPM-Encoder (see Eq. (6) in our paper), but they are not the same thing.
> > > - The output space of our DPM-Encoder is our latent space. Specifically, our latent space is the concatenation of all isometric Gaussian noises $\boldsymbol{\epsilon}_t$ in the denoising process. To use the notation in [2], our latent space is the concatenation of all $d\bar{\boldsymbol{w}}$ in Eq. (6) in [2]. In natural language, DPM-Encoder is about **given a pretrained diffusion model, what noises (in the stochastic denoising process) are more likely to generate this clean image**.
> > > - To align the above two bullet points to our math notations. Eq. (5) in [2] is equivalent to sampling from the posterior distribution $\boldsymbol{x}_\{1:T} \sim q(\boldsymbol{x}_\{1:T}|\boldsymbol{x}_0)$. By contrast, DPM-Encoder samples $\boldsymbol{x}_T \oplus \boldsymbol{\epsilon}_T \oplus \cdots \oplus \boldsymbol{\epsilon}_1 \sim q(\boldsymbol{x}_T \oplus \boldsymbol{\epsilon}_T \oplus \cdots \oplus \boldsymbol{\epsilon}_1|\boldsymbol{x}_0)$.
> > >
> > > > Then, if you freeze the noise during running Eq. (5), like in [1], you get a latent code. I would argue this is equivalent to your DPM-encoder.
> > >
> > > - As we clarify in the response to the above question, the output spaces of Eq. (5) in [2] and DPM-encoder are different. Meanwhile, given a clean image, we think it is unclear how to “freeze the noise during running Eq. (5)”, e.g., freeze the noise as which existing noise? We would like to further discuss if you could clarify it in a follow-up post. We would appreciate it!
> > >
> > >
> > > > In Eq. (10), how do you define $p_\{\boldsymbol{z}}(\boldsymbol{z})$?
> > >
> > > - As discussed at the beginning of Section 3.1, based on our definition of $\boldsymbol{z} := \boldsymbol{x}_T \oplus \boldsymbol{\epsilon}_T \oplus \cdots \oplus \boldsymbol{\epsilon}_1$, the prior distribution $p_\{\boldsymbol{z}}(\boldsymbol{z})$ is the isometric Gaussian distribution, i.e., $p_\{\boldsymbol{z}}(\boldsymbol{z}) = \frac{1}{\sqrt{2 \pi}} \exp \left(-\frac{||\boldsymbol{z}||^2}{2}\right)$. We believed that this question is also related to the misunderstanding regarding our latent space. To be more specific, the prior distribution over our $\boldsymbol{z}$ is simply the isometric Gaussian by definition; by contrast, if one view the noisy images $\boldsymbol{x}_\{1:T}$ as latent codes, then the prior distribution is very complex and depends on the training data.
> > >
> > > > In Appendix F, how do you optimize $\boldsymbol{n}$? As far as I know,  is a long chain and direct back-propagation is nearly impossible.
> > >
> > > - We used fewer discretization steps in this case, in order to backprop efficiently. Note that Appendix F was simply to show an interesting finding that we thought is not worth mentioning in the main text.
> > >
> > > Thanks for raising these questions, which help us clarify the paper. We look forward to your further comments and questions.

---

### Official Review · Reviewer_2RYz · 2022-10-29

**Confidence:** 4
**Correctness:** 3
**Technical Novelty And Significance:** 3
**Empirical Novelty And Significance:** 2
**Recommendation:** 6

**Clarity, Quality, Novelty And Reproducibility:**

Overall, this is a good paper studying an important problem. It demonstrates technically sound formulation and reasonable results. All my concerns are listed above.

The code is not uploaded, but the approach looks simple, the hyperparameters are given in the appendix, but I hope the authors keep the words to make the code public upon acceptance.

**Strength And Weaknesses:**

Strength:
* This paper is mostly well-written with a clear literature review, leading to its motivation.
* The studied problem is important, it is significant to understand the latent space of diffusion models as it has been done in GANs and other generative models. It can lead to more understanding and empirical applications of diffusion models.
* The experimental results empirically support the claims and findings of the paper.

Weaknesses
* Even though the motivation to study the latent space of diffusion models and how this paper studies it with a simple yet effective approach is well-written, the applications to work as a CycleDiffusion, zero-shot image-to-image editing and latent space guidance sounds a little indirect to me, though this may be due to my biased perspective. I directly thought about GAN inversion and related literature, e.g. [1-3], after I read the motivation of this paper. I don't mean the applications in this paper are not okay, but I would suggest making the motivation for the applications tighter.
* As a long-standing problem in understanding the latent space of deep generative models, even though this paper achieves great experimental results, I would still suggest the authors study theoretical guarantees of it.
* The proposed D-CLIP score is a bit confusing, is the relative change meaningful without absolute value?
* Unified Plug-and-Play Guidance is a bit misleading, why would the proposed or defined latent space unify diffusion models and GANs?

[1] Härkönen, E., Hertzmann, A., Lehtinen, J. and Paris, S., 2020. Ganspace: Discovering interpretable gan controls. Advances in Neural Information Processing Systems, 33, pp.9841-9850.
[2] Shen, Y., Yang, C., Tang, X. and Zhou, B., 2020. Interfacegan: Interpreting the disentangled face representation learned by gans. IEEE transactions on pattern analysis and machine intelligence.
[3] Shen, Y. and Zhou, B., 2021. Closed-form factorization of latent semantics in gans. In Proceedings of the IEEE/CVF Conference on Computer Vision and Pattern Recognition (pp. 1532-1540).

**Summary Of The Paper:**

This paper studies the latent space of diffusion models compared to a variety of other generative models such as VAEs, Normalizing Flows, and especially GANs. First of all, this paper provides a very comprehensive and nice overview of latent space across deep generative models and motivates the study of latent space in diffusion models. Then, it traces down to the question about the latent space of stochastic DPMs rather than other methods which leverage "fixed" processes. The authors leverage the finding such that the randomness in the model captures the image-specific information and further utilize the defined latent space for an unpaired image translation task. It explores more tasks such as zero-shot image-to-image editor and latent space guidance with energy-based models.

**Summary Of The Review:**

Overall, my main concern about this paper is the flow of the paper and solid experiments to support the method empirically as theoretical results are missing or challenging. I value the novelty of this paper to formulate this important yet not fully explored problem (latent space of diffusion models). I am leaning towards accepting this paper.

---

> ### Author Response · Authors · 2022-11-13
> **Author response**
>
> Due to response length limitation, we split our response into two parts.

---

> > ### Author Response · Authors · 2022-11-13
> > **Author response (part 1)**
> >
> > > Even though the motivation to study the latent space of diffusion models and how this paper studies it with a simple yet effective approach is well-written, the applications to work as a CycleDiffusion, zero-shot image-to-image editing and latent space guidance sounds a little indirect to me, though this may be due to my biased perspective. I directly thought about GAN inversion and related literature, e.g. [1-3], after I read the motivation of this paper. I don't mean the applications in this paper are not okay, but I would suggest making the motivation for the applications tighter.
> >
> > - Thanks for pointing it out. We would like to clarify the motivations for the applications.
> > - For CycleDiffusion, the main motivation is the limitation of GANs’ (1) generalization and (2) inversion. GANs are typically trained on object-centric images in specific domains, and to the best of our knowledge, there are no GANs that generalize to the large-scale, text-to-image setting. Even if the input at test time is object-centric and domain-specific, it has been widely reported that it is difficult to invert images, with high fidelity, to the latent space of GANs.
> > - For latent space guidance, the main motivation is to show that, in specific domains, different domain experts (GANs, diffusion models, and their variants/hybrids) can be guided in the same formulation. This helps us understand how different models trained on the same data represent sub-populations differently. For instance, it can help us understand the bias and stereotypes in different models by looking at the guided samples.
> >
> > > As a long-standing problem in understanding the latent space of deep generative models, even though this paper achieves great experimental results, I would still suggest the authors study theoretical guarantees of it.
> >
> > - In the updated version, we provide a formal analysis of the *upper bound of image distances* to quantify “similar images”. The complete discussion can be found in the updated paper (at the end of page 4), but we would like to mention it here explicitly. Specifically, we can quantify two intuitions: (1) conditioned on the same text, similar noisy images lead to similar enough denoising directions; (2) Given the same image, two similar texts lead to similar predictions. Given (1) and (2), we show that fixing the latent code (or random seed) gives us a bound of image distance, which does not hold when not fixing the latent code (or random seed).

---

> > > ### Comment · Reviewer_2RYz · 2022-11-27
> > > **Thanks for the clarification**
> > >
> > > Thank the authors for the clarification. I have no further questions. I am leaning towards accepting this paper.

---

> > ### Author Response · Authors · 2022-11-13
> > **Author response (part 2)**
> >
> > > The proposed D-CLIP score is a bit confusing, is the relative change meaningful without absolute value?
> >
> > - The absolute value is shown by the CLIP score, which we have also reported. The motivation for reporting D-CLIP is to measure relative change. Although we used the D-CLIP score as one of the metrics for evaluation, the D-CLIP score itself is not our contribution; prior works have demonstrated that D-CLIP is meaningful to guide or finetune generative models such as StyleGANs [4] [5]. For instance, [5] observed that the CLIP score results in collapsed finetuning of StyleGAN, while the D-CLIP score provides a meaningful training signal. We view it as an *indirect* justification of D-CLIP as a metric.
> >
> > > Unified Plug-and-Play Guidance is a bit misleading, why would the proposed or defined latent space unify diffusion models and GANs?
> >
> > - Thanks for pointing this out. With the defined latent space, (1) the latent prior of various diffusion models and GANs are isometric Gaussian distribution, (2) after a latent code is sampled from this simple Gaussian prior, the synthesis process is deterministic. These two properties allow us to apply plug-and-play guidance, which was originally proposed for GANs [6].
> > - On the other hand, Classifier Guidance, which uses classifiers to guide diffusion models, requires the classifier to be finetuned on noisy images at all noise levels. Because of this finetuning, Classifier Guidance is not a plug-and-play one.
> >
> > > The code is not uploaded, but the approach looks simple, the hyperparameters are given in the appendix, but I hope the authors keep the words to make the code public upon acceptance.
> >
> > - In the updated supplementary material, we included our codes and instructions for all experiments. These will also be open-sourced. After the review process, we will provide an implementation of CycleDiffusion with Stable Diffusion using the Huggingface diffusers library.
> >
> > **Updates of the paper**
> >
> > To avoid a verbatim copy of content, we refer you to our response to all reviewers, titled **Summary of updates**. Here, we briefly summarize the updates.
> >
> > 1. **CycleDiffusion + Cross Attention Control:** We showed that Cross Attention Control (i.e., fixing the cross-attention map) helps CycleDiffusion when the intended structural change is small.
> > 2. **Formal analysis:** In the updated version, we quantify *similar images* as *bound between the distance the two images*. We show that fixing the latent code (or random seed) gives us a bound of image distance, which does not hold when not fixing the latent code (or random seed). This analysis also suggests that more similar texts lead to more similar images when fixing the latent code.
> > 3. **Code uploaded:** Our codes and instructions are now in the supplementary materials. After the review process, we will provide an implementation of CycleDiffusion with Stable Diffusion using the Huggingface diffusers library.
> >
> > We hope that you can consider the updates in the evaluation of this paper. We also look forward to your further comments and questions.
> >
> > [4] StyleCLIP: Textdriven manipulation of StyleGAN imagery. https://arxiv.org/abs/2103.17249
> >
> > [5] StyleGAN-NADA: CLIP-Guided Domain Adaptation of Image Generators. https://arxiv.org/abs/2108.00946
> >
> > [6] Plug & Play Generative Networks: Conditional Iterative Generation of Images in Latent Space. https://arxiv.org/abs/1612.00005

---

### Official Review · Reviewer_NUmH · 2022-10-30

**Confidence:** 5
**Correctness:** 2
**Technical Novelty And Significance:** 1
**Empirical Novelty And Significance:** 2
**Recommendation:** 3

**Clarity, Quality, Novelty And Reproducibility:**


### Clarity

(-) Abbreviation without definition: DPM

> 3.2. DPM-Encoder: an Invertible Encoder ...

> Our contribution is DPM-Encoder, and invertible encoder ...

(-) Considering an existence of a generator as invertibility of an encoder does not make sense. Generally encoder is considered as inverting a generative process.

(-) Broken sentence: several researchers and practitioners have found that when trained with the same "random seed" [missing comma] [missing subject] leads to similar images.

(-) Unclear description: Does above sentence mean that the same $X_T$ and noises leads to similar images across different diffusion models?

(-) Missing definition of "zero-shot". In what aspect is the CycleDiffusion zero-shot?

### Quality (Technical soundness)

(-) Why is generally missing.
* Why is the latent code defined as a concatenation of $X_T$ and intermediate noises?
* Why do the models trained with the same random seed lead to similar images?
* Why does PSNR between a translated image and source image make sense?
* Why does FID between generated and target images make sense? What are generated and target images? In the last sentence of the first paragraph of 4.1 says the target image is the generated image.


(-) Guidance in 3.4 is same with usual guidance in diffusion models.

### Novelty

(-) Latent space = a concatenation of $X_T$ and intermediate noises is trivial.

(-) Encoder = forward process is trivial.

(-) Fixing everything and changing text is trivial.


**Strength And Weaknesses:**


Strengths

(+) This paper shows interesting application of diffusion models.

(+) The proposed latent code (concatenation of all variables) is versatile across different models.


Weaknesses

Please refer to the below section.

**Summary Of The Paper:**

This paper considers a concatenation of all intermediate variables through a forward diffusion as a latent code.

The authors consider the forward process of diffusion models as an encoder.

Using two independently trained diffusion models, the proposed method encodes an image with a fixed random seed for one model and run reverse process on the other model to achieve image-to-image translation.

**Summary Of The Review:**

This paper shows interesting phenomena but does not contribute to building general knowledge with logical development.

---

> ### Author Response · Authors · 2022-11-13
> **Author response**
>
> Due to response length limitation, we split our response into two parts.

---

> > ### Author Response · Authors · 2022-11-13
> > **Author response (part 1)**
> >
> > > Abbreviation without definition: DPM
> > > Missing definition of "zero-shot". In what aspect is the CycleDiffusion zero-shot?
> >
> > - Thanks for pointing that out. DPM stands for diffusion probabilistic model, including models with a fixed forward process and an explicit probabilistic formulation (to distinguish from DDGAN and DiffAE). By zero-shot, we mean the model has never seen the text pairs and image pairs (used for inference) during training. We have clarified them in the updated version.
> >
> > > Considering the existence of a generator as the invertibility of an encoder does not make sense. Generally, an encoder is considered as inverting a generative process.
> >
> > - We have changed “invertibility” to “perfect reconstruction” to avoid ambiguity.
> >
> > > Broken sentence: several researchers and practitioners have found that when trained with the same "random seed" [missing comma] [missing subject] leads to similar images.
> > > Unclear description: Does the above sentence mean that the same $\boldsymbol{x}_T$ and noises lead to similar images across different diffusion models?
> >
> > - Yes, this sentence should be “several researchers and practitioners have found that sampling with the same "random seed" leads to similar images”. In the updated version, we also provide a formal analysis of the *upper bound of image distances* to quantify “similar images”. The complete discussion can be found in the updated paper (at the end of page 4), but we would like to mention it here explicitly. Specifically, we can quantify two intuitions: (1) conditioned on the same text, similar noisy images lead to similar enough denoising directions; (2) Given the same image, two similar texts lead to similar predictions. Given (1) and (2), we show that fixing the latent code (or random seed) gives us a bound of image distance, which does not hold when not fixing the latent code (or random seed).

---

> > ### Author Response · Authors · 2022-11-13
> > **Author response (part 2)**
> >
> > > Why is the latent code defined as a concatenation of $\boldsymbol{x}_T$ and intermediate noises?
> >
> > - As we explained in the introduction and Section 3.1, we want to define the image synthesis process as a deterministic mapping from latent codes to images.  Moreover, we want to achieve a perfect reconstruction and preserve the dynamics of the noise. Defining the latent code as the concatenation of $\boldsymbol{x}_T$ and intermediate noises satisfies this goal.
> >
> > > Why do the models trained with the same random seed lead to similar images?
> >
> > - As clarified above, the observation is: sampling with the same "random seed" leads to similar images. The models can be trained with different initializations and model architectures, as long as the two models share a forward process. In the updated version, we further provide a formal analysis. See our answer to the question starting with “Broken sentence”.
> >
> > > Why does PSNR between a translated image and a source image make sense?
> > > Why does FID between generated and target images make sense?
> >
> > - Both of these follow previous works on unpaired image-to-image translation, e.g., the EGSDE baseline. To make it more explicit, PSNR and SSIM show the minimality of changes; FID shows the closeness to the target distribution. No metric is perfect, but these are the best practices we can follow.
> >
> > > What are generated and target images? The last sentence of the first paragraph of 4.1 says the target image is the generated image.
> >
> > - Suppose the goal is to translate a set of cat images into dog images. Generated images are the ones generated by each method. Target images are real dog images in AFHQ Dog. We have clarified it in the updated version.
> >
> > > Guidance in 3.4 is the same as the usual guidance in diffusion models.
> >
> > - We respectfully **disagree** with this claim. In classifier guidance (which we believe is what “usual guidance” means here), the classifier is finetuned at all levels of noisy images and applied to each noise level sequentially during sampling. In plug-and-play guidance in Section 3.4, the classifier can be any pre-trained classifier on clean images, and all noises are updated simultaneously. Plug-and-play guidance avoids the need to finetune the classifier.
> >
> >
> > Finally, we like the characterization of our proposed method as "simple" or "trivial" since we favor such straightforward solutions that work. We even pointed out in our paper that this formulation of latent code is by definition (in other words, “simple” or “trivial”). However, we would like to clarify two main contributions of this paper.
> >
> > 1. Our first contribution is DPM-Encoder, which makes it explicit how to sample latent codes from real images (i.e., a posterior distribution of latent codes) using stochastic diffusion models. For us, this is also simple or trivial, but it is very useful (as we show later) and to the best of our knowledge no one has proposed it before.
> > 2. Our second contribution is to suggest two interesting applications: CycleDiffusion and plug-and-play guidance. More updated contributions such as a more formal analysis of the fact that fixed random noises lead to similar images, and the combination of CycleDiffusion and Cross Attention Control) are summarized below.
> >
> >
> > **Updates of the paper**
> >
> > To avoid a verbatim copy of content, we refer you to our response to all reviewers, titled **Summary of updates**. Here, we briefly summarize the updates.
> >
> > 1. **CycleDiffusion + Cross Attention Control:** We showed that Cross Attention Control (i.e., fixing the cross-attention map) helps CycleDiffusion when the intended structural change is small.
> > 2. **Formal analysis:** In the updated version, we quantify *similar images* as *bound between the distance the two images*. We show that, by fixing the latent code, we can get tighter bounds than not fixing the latent code. This analysis also suggests that more similar texts lead to more similar images when fixing the latent code.
> > 3. **Code uploaded:** Our codes and instructions are now in the supplementary materials. After the review process, we will provide an implementation of CycleDiffusion with Stable Diffusion using the Huggingface diffusers library.
> >
> > We hope that you can consider the updates in the evaluation of this paper. We also look forward to your further comments and questions.

---

> > > ### Comment · Reviewer_NUmH · 2022-11-27
> > > **Thanks**
> > >
> > > Thank you for the response.
> > >
> > > The 1st, 4th, and 5th concerns are resolved.
> > > Though PSNR and FID are not quite supported, I will consider them as difficulties that exist in the literature.
> > >
> > > I carefully checked eq. 7-9 but they are not enough.
> > > - Assumptions 1 and 2 are sloppy because $K_t$ and $S_t$ are not bounded.
> > > - $B^s_T$ is not defined. Maybe $B^s_T=||x_T-\hat{x}_T||_2$?
> > > - Why $||x_T-\hat{x}_t||_2=0$ is not provided. It is not straightforward. Maybe a typo of $||x_T-\hat{x}_T||_2=0$? It fixes above two flaws except the superscript $s$.
> > >
> > > minor:
> > > - Timestep $t$ and text $\mathbf{t}$ are confusing.
> > >
> > > Furthermore, I also prefer simple methods than complex ones but they should be supported by some principle. On the other hand, trivial methods do not provide any principle or generalizable knowledge to the research community and can be done by anyone.
> > >
> > > In my opinion, the proposed method is interesting but it is not as rigorously grounded as the ICLR standard, i.e., the paper provides a working method but why it works is not clear. I expect the next version to reinforce the link between the two diffusion models through the shared latent space.
> > >
> > > I keep my original rating.

---

> > > > ### Author Response · Authors · 2022-12-11
> > > > **Thanks for you comments**
> > > >
> > > > We are glad to hear that most technical questions have been addressed. For Eq. (7) - Eq. (9), it should be $||x_T - \hat{x}_T||_2 = 0$, and $B_t$ is the upper bound of $||x_t - \hat{x}_t||_2$. Since the distinction between "simple" and "trivial" is somewhat subjective, we would refrain from further commenting on this topic.

---

> > ### Author Response · Authors · 2022-11-20
> > **Follow-up with Reviewer NUmH**
> >
> > Thanks for your valuable comments. We are writing this follow-up post because it has been days since we posted our initial response. Please let us know if your initial concerns are addressed, and we would appreciate any further questions or comments.

---

### Official Review · Reviewer_JmnA · 2022-11-01

**Confidence:** 2
**Correctness:** 3
**Technical Novelty And Significance:** 2
**Empirical Novelty And Significance:** 2
**Recommendation:** 5

**Clarity, Quality, Novelty And Reproducibility:**

In my opinion, the clarity of the paper could be improved. I find that many sentences are vague e.g. " a common latent space 'emerges' from...". It could be more streamlined and direct what the point is.

In terms of novelty - the main contribution seems to be a way of defining a Gaussian latent space + DPM-Encoder for diffusion model and to exploit this perspective for unpaired image translation, zero shot image-to-image editing and conditioning the generation.

In terms of reproducibility, code is not provided in the submission but details of experimental setup are somewhat comprehensive.


**Strength And Weaknesses:**

Section 3.2: do I understand correctly that your DPM-Encoder is a proposal for how to build a z such that x = G(z) for z ~ DPMEnc(z | x, G), which amount to just concatenating x_T with \eps_t as described?

Can you clarify the following: in section 3.1 I understood that in your unified framework, z is drawn from a Gaussian. In equation (6), it doesn't seem to be the case. In section 3.2, is the Gaussian latent  abandoned?

I am a bit confused about the proof in Proposition 1. Do you use induction ((show it’s true for 0, assume it’s true for t, show it’s true for t+1)
) or backward induction (show it’s true for T, assume it’s true for t, show it’s true for t-1)  ?
Also,  if the goal here to show that G(z) = x for any z sampled from DPMEnc ( | x), in the statement I would write x = G(z) =: \bar{x}, just for clarity.

Section 3.3:
Can you clarify what you mean with “our defined latent codes contain all randomness?”
“ we can formalize the above finding as a common latent space emerges..." :can you clarify this sentence (what does it mean pragmatically?) ?
How is the concept of  ‘common’ defined and measured?

What do you mean “D_1 and D_2 do not need to be two datasets?”

Section  4.1
Peak Signal-to-Noise Ratio: what is this an indicator of? Meaning CycleDiffusion has excellent  SSIM (which  is more correlated to human perception) than baselines, but lower PSNR (actually the lowest). What is this an indicator of?

 Section 4.3
Am I correct to understand that your contribution in section 4.3 is the fact that you can “work on a latent space” for diffusion models (e.g. apply Langevin) similarly to what one can do with GANs?
The findings reported are interesting but reporting them without a critical explanation is maybe limiting. E.g “3D GANs are worse at guidance than 2D GANs and diffusion models”: why so? Or at least, what’s your opinion on this?




**Summary Of The Paper:**

The paper provides a unified setting among generative models. In particular it offers a perspective  on diffusion models (DM) that cast them in a framework more similar to GANs, which admit a Gaussian formulation of the latent space. This point of view is then leveraged in three settings.
Contributions: 1. provide a formulation of DM with Gaussian latent code  (introducing the concept of DPM-encoder that maps images into a latent space). 2. Propose CycleDiffusion for a) image-to-image translation and b) to show that text-to-image diffusion models can be used as zero-shot image-to-image editors. 3. Rely on the proposed perspective which unifies Diffusion Models and GANs to perform plug-and-play guidance.

**Summary Of The Review:**

I think the paper could improve in its clarity. The novelty seems pretty limited, although CycleDiffusion used as zero-shot image-to-image editor is interesting in my opinion.

---

> ### Author Response · Authors · 2022-11-13
> **Author response**
>
> Due to response length limitation, we split our response into two parts.

---

> > ### Author Response · Authors · 2022-11-13
> > **Author response (part 1)**
> >
> > > Section 3.2: do I understand correctly that your DPM-Encoder is a proposal for how to build a z such that x = G(z) for z ~ DPMEnc(z | x, G), which amounts to just concatenating x_T with \eps_t as described?
> >
> > - The concatenation $\boldsymbol{x}_T \oplus \boldsymbol{\epsilon}_T \oplus \cdots \oplus \boldsymbol{\epsilon}_1$ is our definition of the Gaussian latent code $\boldsymbol{z}$. DPM-Encoder is a proposal for how to sample $\boldsymbol{z}$ from a given real image $\boldsymbol{x}$, such that $\boldsymbol{x} = G(\boldsymbol{z})$.
> >
> > > Can you clarify the following: in section 3.1 I understood that in your unified framework, z is drawn from a Gaussian. In equation (6), it doesn't seem to be the case. In section 3.2, is the Gaussian latent abandoned?
> >
> > - Both Section 3.1 and Section 3.2 use the same definition of latent code $\boldsymbol{z} := \boldsymbol{x}_T \oplus \boldsymbol{\epsilon}_T \oplus \cdots \oplus \boldsymbol{\epsilon}_1$. They are two sides of the same problem, detailed as follows.
> > - Section 3.1 is about image synthesis (in other words, decoding) $\boldsymbol{x} = G(\boldsymbol{z})$, i.e., how to map latent code $\boldsymbol{z}$ to image $\boldsymbol{x}$. To sample a set of images, we sample $\boldsymbol{z}$ from the **prior** Gaussian and then map them to images.
> > - Section 3.2 is about encoding $\boldsymbol{z} \sim \text{DPM-Encoder}(\boldsymbol{z}|\boldsymbol{x}, G)$, i.e., how to sample $\boldsymbol{z}$ based on a given real image $\boldsymbol{x}$ such that $\boldsymbol{x} = G(\boldsymbol{z})$. When the neural network exactly models the score function, sampling from DPM-Encoder means sampling from the **posterior** $q(\boldsymbol{x}_T \oplus \boldsymbol{\epsilon}_T \oplus \cdots \oplus \boldsymbol{\epsilon}_1|\boldsymbol{x}_0)$.
> >
> >
> > > I am a bit confused about the proof in Proposition 1. Do you use induction ((show it’s true for 0, assume it’s true for t, show it’s true for t+1) ) or backward induction (show it’s true for T, assume it’s true for t, show it’s true for t-1)? Also, if the goal here is to show that G(z) = x for any z sampled from DPMEnc ( | x), in the statement I would write x = G(z) =: \bar{x}, just for clarity.
> >
> > - We are using the latter one (show it’s true for T, assume it’s true for t, show it’s true for t-1). Thanks for pointing this out! We have updated it.
> >
> > > Section 3.3: Can you clarify what you mean by “our defined latent codes contain all randomness?” “We can formalize the above finding as a common latent space emerges..." : can you clarify this sentence (what does it mean pragmatically?)?
> > How is the concept of ‘common’ defined and measured?
> >
> > - In the updated version, we provide a more formal description. Specifically, we provide a formal analysis of the *upper bound of image distances* to quantify “similar images”. The complete discussion can be found in the updated paper (at the end of page 4), but we would like to mention it here explicitly. Specifically, we can quantify two intuitions: (1) conditioned on the same text, similar noisy images lead to similar enough denoising directions; (2) Given the same image, two similar texts lead to similar predictions. Given (1) and (2), we show that fixing the latent code (or random seed) gives us a bound of image distance, which does not hold when not fixing the latent code (or random seed).
> >
> > > What do you mean by “D_1 and D_2 do not need to be two datasets?”
> >
> > - This means that D_1 and D_2 are basically two distributions of images. One may say that a dataset is a distribution (e.g., a cat dataset, a dog dataset); one may also say that a text condition is a distribution (e.g., all images captioned by “a photo of a cat”, all images captioned by “a photo of a dog”). In the updated version, we have also clarified this.
> >
> > > Section 4.1 Peak Signal-to-Noise Ratio: what is this an indicator of? Meaning CycleDiffusion has excellent SSIM (which is more correlated to human perception) than baselines, but lower PSNR (actually the lowest). What is this an indicator of?
> >
> > - We followed the EGSDE paper to report FID, PSNR, and SSIM. Intuitively, both PSNR and SSIM are indicators for content preservation, or in other words, how similar the generated image looks to the original one. However, even if among the baselines, these two metrics are not consistent. For instance, the GAN-based CUT method achieves much higher SSIM than diffusion-based methods, while its PSNR is generally lower than those of diffusion models.
> > - In the updated version, we have also tuned the number of encoding steps to improve the performance of CycleDiffusion on unpaired image-to-image translation. Both PSNR and SSIM are improved, and PSNR is comparable to or higher than some baselines now.

---

> > > ### Comment · Reviewer_JmnA · 2022-11-22
> > > **Thank for the response**
> > >
> > > I thank the authors for their response and for clarifying a few aspects! I find the explanation at the end of page 4) a bit obscure and the formalization stretched - it doesn't seem to add much to the intuition in my opinion.
> > > Given the combination of strengths and weaknesses, I'll  keep my score unchanged.

---

> > ### Author Response · Authors · 2022-11-13
> > **Author response (part 2)**
> >
> > > Section 4.3 Am I correct to understand that your contribution in section 4.3 is the fact that you can “work on a latent space” for diffusion models (e.g. apply Langevin) similarly to what one can do with GANs? The findings reported are interesting but reporting them without a critical explanation is maybe limiting. E.g “3D GANs are worse at guidance than 2D GANs and diffusion models”: why so? Or at least, what’s your opinion on this?
> >
> > - The motivation of this part is to show that different variants of diffusion models and GANs can be conditioned/guided in the same way. In the updated version, we put more focus on the fact that *different models behave differently even guided by the same model* instead of discussing *which model is better*.
> >
> > > In my opinion, the clarity of the paper could be improved. I find that many sentences are vague e.g. " a common latent space 'emerges' from...". It could be more streamlined and direct what the point is.
> > > In terms of novelty - the main contribution seems to be a way of defining a Gaussian latent space + DPM-Encoder for diffusion model and exploiting this perspective for unpaired image translation, zero-shot image-to-image editing, and conditioning the generation.
> >
> > - In the updated version, we provide a more formal description. Specifically, we provide a formal analysis of the *upper bound of image distances* to quantify “similar images”. Please see our answer to the question starting with “Section 3.3: Can you clarify …”. We are glad that the reviewer acknowledges the value in the application of this technique to image editing from the text.
> >
> > > In terms of reproducibility, code is not provided in the submission but details of the experimental setup are somewhat comprehensive.
> >
> > - In the updated supplementary material, we included our codes and instructions for all experiments. These will also be open-sourced. Further, we will provide an implementation of CycleDiffusion with Stable Diffusion using the Huggingface diffusers library.
> >
> > **Updates of the paper**
> >
> > To avoid a verbatim copy of content, we refer you to our response to all reviewers, titled **Summary of updates**. Here, we briefly summarize the updates.
> >
> > 1. **CycleDiffusion + Cross Attention Control:** We showed that Cross Attention Control (i.e., fixing the cross-attention map) helps CycleDiffusion when the intended structural change is small.
> > 2. **Formal analysis:** In the updated version, we quantify *similar images* as *bound between the distance the two images*. We show that fixing the latent code (or random seed) gives us a bound of image distance, which does not hold when not fixing the latent code (or random seed). This analysis also suggests that more similar texts lead to more similar images when fixing the latent code.
> > 3. **Code uploaded:** Our codes and instructions are now in the supplementary materials. After the review process, we will provide an implementation of CycleDiffusion with Stable Diffusion using the Huggingface diffusers library.
> >
> >
> > We hope that you can consider the updates in the evaluation of this paper. We also look forward to your further comments and questions.

---

> > ### Author Response · Authors · 2022-11-20
> > **Follow-up with Reviewer JmnA**
> >
> > Thanks for your valuable comments. We are writing this follow-up post because it has been days since we posted our initial response. Please let us know if your initial concerns are addressed, and we would appreciate any further questions or comments.

---

### Official Review · Reviewer_Nh8B · 2022-11-01

**Confidence:** 3
**Clarity, Quality, Novelty And Reproducibility:** 1. While the proposed stochastic enco…
**Correctness:** 3
**Technical Novelty And Significance:** 3
**Empirical Novelty And Significance:** 4
**Recommendation:** 8

**Strength And Weaknesses:**

1. The idea of defining the stochastic encoder is simple but interesting.
2. Being able to encode an image and using the code for image to image translation or editing is clever.
3. The empirical results on both image to image translation and text guided edits are impressive.

**Summary Of The Paper:**

The paper proposes a way of deriving an encoder from a pretrained diffusion model. In particular, this encoder is stochastic and invertible, which makes it different from existing attempts. The authors show that this encoder can be deployed in an image to image translation setting, and demonstrates interesting results. Moreover, a unified guidance method is proposed based on the derived latent representations.

**Summary Of The Review:**

Overall I think this is a promising paper with a simple idea and interesting results. I'd like to hear clarifications on my questions during the rebuttal phase.

---

> ### Author Response · Authors · 2022-11-13
> **Author response**
>
> > While the proposed stochastic encoder is interesting, I wonder how it compares with the ODE-based deterministic encoder empirically. Will the stochasticity give any benefits to either the results or computational cost over the deterministic version?
>
> -  Thanks for the question. Yes, we have this experiment, and here we make it more explicit. As we discuss in the related work, the connection between CycleDiffusion and the DDIB baseline (in Table 4) is: DDIB uses the ODE-based deterministic encoder to deterministically encode real images into a latent code; CycleDiffusion uses our DPM-Encoder to stochastically sample latent codes based on the real images. Results in Table 4 show that CycleDiffusion greatly outperformed DDIB in terms of image editing. However, the computational cost is not reduced.
>
> > I have found that the plug-and-play guidance part is rather disjoint from the main contribution and not properly justified. My understanding of it is that the authors propose to formulate a text-conditioned energy function using a CLIP model. One can then guide a given generative model by optimizing its latent, which applies to both GANs and diffusion models. My question is that how is this related to the stochastic encoder? Why can't you just optimize the initial noise $\boldsymbol{x}_T$ following a DDIM inference procedure? Maybe I'm missing something but I think it'd be good to clarify this.
>
> - This is correct. This experiment is not related to DPM-Encoder since no real images are involved here (similarly, the ODE-based deterministic encoder is not used here either).
> - “Optimize the initial noise $\boldsymbol{x}_T$ following a DDIM inference procedure” is done in LDM-DDIM (Figure 6 in the updated version).
> - The goal of this guidance experiment is to show that diffusion models (stochastic, deterministic, DDGAN, DiffAE) can be guided in the same way as GANs when the latent space is defined as isometric Gaussian. The connection between the plug-and-play guidance and CycleDiffusion is that they are both consequences of the Gaussian latent space for diffusion models. However, we agree that the plug-and-play guidance part is less interesting than the CycleDiffusion part. In the updated version, we provide more analysis for CycleDiffusion and reduce the space for the plug-and-play guidance.  We have clarified this in the paper and detailed updates are provided as follows.
>
> **Updates of the paper**
>
> To avoid a verbatim copy of content, we refer you to our response to all reviewers, titled **Summary of updates**. Here, we briefly summarize the updates.
>
> 1. **CycleDiffusion + Cross Attention Control:** We showed that Cross Attention Control (i.e., fixing the cross-attention map) helps CycleDiffusion when the intended structural change is small.
> 2. **Formal analysis:** In the updated version, we quantify *similar images* as *bound between the distance the two images*. We show that fixing the latent code (or random seed) gives us a bound of image distance, which does not hold when not fixing the latent code (or random seed). This analysis also suggests that more similar texts lead to more similar images when fixing the latent code.
> 3. **Code uploaded:** Our codes and instructions are now in the supplementary materials. After the review process, we will provide an implementation of CycleDiffusion with Stable Diffusion using the Huggingface diffusers library.
>
> We hope that you can consider the updates in the evaluation of this paper. We also look forward to your further comments and questions.

---

> > ### Comment · Reviewer_Nh8B · 2022-11-18
> > **Re**
> >
> > Thanks to the authors for the clarifications. My major concerns are addressed, although I still believe that it'd be beneficial to focus and elaborate more on the real image editing part and maybe move the plug and play guidance part completely to the appendix. I'm raising my score to 8.

---

> > > ### Author Response · Authors · 2022-11-18
> > > **Thank you for your feedback**
> > >
> > > Thank you for acknowledging our efforts in addressing the concerns. The updated version reduces the space of plug-and-play guidance to accommodate more contents on real image editing (i.e., Updates 1 and 2). Since Discussion Stage 1 is coming to an end, we will update the paper in the next version to consider comments from all reviewers. Please let us know if you have further questions or comments.

---

### Official Review · Reviewer_dmqB · 2022-11-03

**Confidence:** 4
**Correctness:** 2
**Technical Novelty And Significance:** 2
**Empirical Novelty And Significance:** 2
**Recommendation:** 6

**Clarity, Quality, Novelty And Reproducibility:**

The paper is over-all well-written. The results are novel and original.

The authors propose some new ideas (usefulness of the forward dynamic in context of stochastic models) and explicitly state the hypothesis that the diffusion models with similar noise schedule share latent spaces.

 The practical quality of the paper is good since the authors provide a broad comparison of their method with existing approaches. One question: How do ILVR, SDEdit, EGSDE models apply to the problem of unpaired image-to-image translation (Cat vs Dog and Wild vs Dog)? As I understand, the mentioned methods are not directly aimed at the unpaired im-to-im translation by design.

 The code is not provided in the supplementary materials.


**Strength And Weaknesses:**

Strength:
1) The proposed idea is clear and practically applicable. One can easily apply the proposed framework for a general stochastic diffusion model.

2) The experiments look quite convincing.

Weaknesses:
1) The idea of DPM-Encoder itself seems to be quite easy and straightforward. All of the papers dealing with stochastic diffusion models (like DDPM) actually mentioned the forward form of the stochastic dynamic (i.e. how to obtain latent code $x_T$ from the original image $x_0$) but didn’t stress the importance of the dynamic. Therefore, the only theoretical contribution of the paper under consideration is that the authors highlight the importance of forward stochastic dynamics (for the problem of unpaired image-to-image generation related applications)

2) As it even mentioned in the article itself, the methodology of stochastic diffusion based unpaired image-to-image generation is under question and requires theoretical verification.

**Summary Of The Paper:**

The paper under consideration proposes a method for inverting stochastic diffusion models. Based on this method the authors come up with an (conditional) unpaired image-to-image translation approach called “CycleDiffusion”. The validity of the proposed methodology is verified by several applications.

**Summary Of The Review:**

The authors provide a new method for solving (conditional) unpaired im-to-im problems based on stochastic diffusion models. In spite of the interesting methodological proposition and good practical results, the method requires further theoretical investigation.

---

> ### Author Response · Authors · 2022-11-13
> **Author response**
>
> > The idea of DPM-Encoder itself seems to be quite easy and straightforward. All of the papers dealing with stochastic diffusion models (like DDPM) actually mentioned the forward form of the stochastic dynamic (i.e. how to obtain latent code $\boldsymbol{x}_T$ from the original image $\boldsymbol{x}_0$) but didn’t stress the importance of the dynamic. Therefore, the only theoretical contribution of the paper under consideration is that the authors highlight the importance of forward stochastic dynamics (for the problem of unpaired image-to-image generation related applications)
>
> - Thanks for the thoughtful summary, and we have a few additional comments.
> - We think this paper is stressing the importance of the reverse process (i.e., the denoising process) dynamics. In a stochastic diffusion model, $\boldsymbol{x}_t \sim \mathcal{N}(\sqrt{\bar{\alpha}_t}\boldsymbol{x}_0, (1 - \bar{\alpha}_t)\boldsymbol{I})$; therefore, we argue that simply using an intermediate $\boldsymbol{x}_t$ as a latent code will lose details in $\boldsymbol{x}_0$.
>
> - We agree that DPM-Encoder is simple to build, and it samples $\boldsymbol{x}_T \oplus \boldsymbol{\epsilon}_T \oplus \cdots \oplus \boldsymbol{\epsilon}_1$ based on $\boldsymbol{x}_0$. To compute the noise $\boldsymbol{\epsilon}_t$ at each step, we first sample two consecutive noisy images $\boldsymbol{x}_t$ and $\boldsymbol{x}_\{t-1}$ from the forward process, and then we compute the noise $\boldsymbol{\epsilon}_t$ based on $\boldsymbol{x}_t$ and $\boldsymbol{x}_\{t-1}$ by definition. When the neural network exactly models the score function, sampling from DPM-Encoder means sampling from the posterior $q(\boldsymbol{x}_T \oplus \boldsymbol{\epsilon}_T \oplus \cdots \oplus \boldsymbol{\epsilon}_1|\boldsymbol{x}_0)$.
>
> > As it even mentioned in the article itself, the methodology of stochastic diffusion-based unpaired image-to-image generation is under question and requires theoretical verification.
>
> - In the updated version, we provide a formal analysis of the *upper bound of image distances* to quantify “similar images”. The complete discussion can be found in the updated paper (at the end of page 4), but we would like to mention it here explicitly. Specifically, we try to quantify two intuitions: (1) conditioned on the same text, similar noisy images lead to similar enough denoising directions; (2) Given the same image, two similar texts lead to similar predictions. Given (1) and (2), we show that fixing the latent code (or random seed) gives us a bound of image distance, which does not hold when not fixing the latent code (or random seed).
>
> > One question: How do ILVR, SDEdit, and EGSDE models apply to the problem of unpaired image-to-image translation (Cat vs Dog and Wild vs Dog)? As I understand, the mentioned methods are not directly aimed at the unpaired image-to-image translation by design.
>
> - For the unpaired experiments, we follow the experimental setting in EGSDE. EGSDE is directly aimed at the unpaired image-to-image translation, and we copied the numbers of ILVR and SDEdit reported in this paper. To make it explicit, for ILVR, the original images are processed with a low-pass filter and used as the condition for the target domain diffusion model; for SDEdit, noises are added to the original images, and the noisy image is denoised with the target domain diffusion model.
>
> > The code is not provided in the supplementary materials.
>
> - In the updated supplementary material, we included our codes and instructions for all experiments. These will also be open-sourced. After the review process, we will provide an implementation of CycleDiffusion with Stable Diffusion using the Huggingface diffusers library.
>
> **Updates of the paper**
>
> To avoid a verbatim copy of content, we refer you to our response to all reviewers, titled **Summary of updates**. Here, we briefly summarize the updates.
>
> 1. **CycleDiffusion + Cross Attention Control:** We showed that Cross Attention Control (i.e., fixing the cross-attention map) helps CycleDiffusion when the intended structural change is small.
> 2. **Formal analysis:** In the updated version, we quantify *similar images* as *bound between the distance the two images*. We show that fixing the latent code (or random seed) gives us a bound of image distance, which does not hold when not fixing the latent code (or random seed).  This analysis also suggests that more similar texts lead to more similar images when fixing the latent code.
> 3. **Code uploaded:** Our codes and instructions are now in the supplementary materials. After the review process, we will provide an implementation of CycleDiffusion with Stable Diffusion using the Huggingface diffusers library.
>
> We hope that you can consider the updates in the evaluation of this paper. We also look forward to your further comments and questions.

---

### Author Response · Authors · 2022-11-13
**Summary of updates**

We thank all six reviewers for their valuable feedback. In the updated version, we made major changes that are summarized below:

**Update 1: CycleDiffusion + Cross Attention Control**

The core observation behind CycleDiffusion is that fixed random noises lead to similar images. CycleDiffusion infers and fixes these random noises for image editing. A recent work [1] found that fixing the cross-attention map (i.e., Cross Attention Control or CAC) further improves the minimality of the difference between generated images.

In **Section 4.2 of the updated version**, we show that CAC can be applied to CycleDiffusion. For instance, **Figure 4 of the updated version** shows CAC helps CycleDiffusion when the intended structural change is small.

[1] Prompt-to-Prompt Image Editing with Cross Attention Control


**Update 2: formal analysis**

A common question raised by reviewers is *why does fixing the latent code give us similar images when the model is conditioned on two similar texts*. In **Section 3.3 of the updated version**, we quantify *similar images* as the *bounded distance between the two images*. Specifically, we show that fixing the latent code (or random seed) gives us a bounded distance between images, which does not hold when not fixing the latent code (or random seed).  This analysis also suggests that more similar texts lead to more similar images when fixing the latent code.


**Update 3: code uploaded**

Please find our codes and instructions to run them in the supplementary materials. After the review process, we will provide an implementation of CycleDiffusion with Stable Diffusion using the Huggingface diffusers library.

---

### Decision · Program_Chairs · 2023-01-20

**Decision:**

Reject

**Justification For Why Not Higher Score:**

The presentation and the need for and execution of the unification is unclear and clunky.

**Justification For Why Not Lower Score:**

N/A

**Metareview: Summary, Strengths And Weaknesses:**

The paper propose a unification to the latent space of diffusion models. This unification is then used to methods like a cycle diffusion. The idea for a latent space for stochastic diffusion probabilistic models is to concatenate all the noises used to diffuse together into a single vector. Keeping these noises level fixed in another generator allows for images to be translated from one distribution to another. That is, cycle generation. Overall, the performance is intriguing, though the presentation and the need for and execution of the unification is unclear and clunky. The presentation really led to the difference in opinions between reviewers. For example, after reading the author reply, reviewer JmnA writes

" I find the explanation at the end of page 4) a bit obscure and the formalization stretched - it doesn't seem to add much to the intuition in my opinion,"

which is supported by reviewer NUmH in reply to the authors

"I also prefer simple methods than complex ones but they should be supported by some principle."

The results are nice, but the entire presentation could be simplified to only stochastic diffusion probabilistic models and enriched by going into more detail on why the concatenation is the correct choice with the extra page or so of space that would be freed.